# Accurate global and local 3D alignment of cryo-EM density maps using local spatial structural features

Bintao He [1,5], Fa Zhang[2,5], Chenjie Feng[3], Jianyi Yang [1], Xin Gao [4] ✉ & Renmin Han [1] ✉

Advances in cryo-electron microscopy (cryo-EM) imaging technologies have led to a rapidly increasing number of cryo-EM density maps. Alignment and comparison of density maps play a crucial role in interpreting structural information, such as conformational heterogeneity analysis using global alignment and atomic model assembly through local alignment. Here, we present a fast and accurate global and local cryo-EM density map alignment method called CryoAlign, that leverages local density feature descriptors to capture spatial structure similarities. CryoAlign is a feature-based cryo-EM map alignment tool, in which the employment of feature-based architecture enables the rapid establishment of point pair correspondences and robust estimation of alignment parameters. Extensive experimental evaluations demonstrate the superiority of CryoAlign over the existing methods in terms of both alignment accuracy and speed.

Density maps obtained through cryo-electron microscopy (cryo-EM) provide key information for protein structure determination and function analysis[1,2]. The Electron Microscopy Data Bank[3], a public database, has accumulated more than thirty thousand entries as of October 2023, with a fourfold increase since 2018. Moreover, with the advancement of cryo-EM technology, most of the recently solved cryo-EM structures have high resolution, ranging from 2 Å to 10 Å. Many important works[4–6] explore the continuous conformation changes to reconstruct a series of high-resolution maps, sufficiently enriching and characterizing the landscape of molecular states. All these factors indicate the coming of a high-resolution and big-data cryo-EM era. To extract and interpret the underlying structural information from cryo-EM density maps, there is a strong demand for accurate alignment and comparison of cryo-EM maps, especially for entries with high resolution. For example, comparison of superimposed density maps helps to identify variable areas associated with heterogeneity and to integrate 3D classification to establish conformational landscapes[7–13]. In protein

macromolecular complex modeling, accurate local alignment effectively accelerates the chain assembly process[14–17], as the density of a subunit structure is simulated to find the best matching regions in experimental maps[18–21]. Additionally, similarity scores derived from alignment can serve as feasible metrics for cryo-EM map retrieval problems[22,23]. However, density maps with high and medium resolutions contain a substantial amount of rich and clear structural information, placing high requirements on alignment accuracy and efficiency.

Several works have been developed to address the cryo-EM map alignment problem. gmfit[24,25] represents cryo-EM density maps with Gaussian mixture models (GMM) and utilizes maximization of the correlation between Gaussian functions to optimize the global transformation parameters. The balance between speed and approximation accuracy of GMM is determined by the number of Gaussian kernels used. gmfit utilizes a combination of Gaussian functions far less than the total number of raw atoms to represent a map, providing fast and

[1]Research Center for Mathematics and Interdisciplinary Sciences, Shandong University, Qingdao 266237, China. [2]School of Medical Technology, Beijing Institute of Technology, Beijing 100081, China. [3]College of Medical Information and Engineering, Ningxia Medical University, Yinchuan 750004, China. [4]King Abdullah University of Science and Technology (KAUST), Computational Bioscience Research Center (CBRC), Computer, Electrical and Mathematical Sciences and Engineering (CEMSE) Division, Thuwal 23955, Saudi Arabia. [5]These authors contributed equally: Bintao He, Fa Zhang. ✉e-mail: xin.gao@kaust.edu.sa; hanrenmin@sdu.edu.cn

robust, but less accurate alignment results, which makes gmfit suitable for low-resolution maps. Chimera, a widely used software for molecular manipulation and visualization, offers a map fitting method known as fitmap[26]. fitmap directly performs local optimization to maximize the correlation between voxels, starting from multiple randomly generated initial placements of the source map. However, due to the significant influence of the initial location of the maps, fitmap typically requires users' intervention or the use of preset locations to achieve satisfactory results. Recently, a vector-based cryo-EM density map alignment method called VESPER[27] was proposed for better alignment and retrieval performance. VESPER utilizes a collection of vectors that are specifically oriented toward the local density maximum to capture the intricate 3D structures embedded in the maps[28]. Using the sum of dot products between matched vectors from two maps, VESPER finds the best alignment parameters via exhaustive search of both rotational and translational intervals. Compared to gmfit and fitmap, the point distribution retains abundant information about spatial structures and the orientations of vectors explicitly depict local density trends. However, the parameter optimization of VESPER is based on an exhaustive search on spatial rotation and translation with a given search interval, which leads to inflexible and insufficient optimization and considerable execution time.

Here, we propose a global and local cryo-EM density map alignment method, CryoAlign, to achieve fast, accurate and robust comparison of two cryo-EM density maps by utilizing local spatial feature descriptors. In CryoAlign, the density map is sampled to generate a point cloud representation, and a clustering process is applied to the point cloud to extract key points based on local properties such as the density value distribution and the connectivity of points. Once the key points are identified, CryoAlign calculates local feature descriptors by collecting the distribution of density directions in their vicinity. These feature descriptors capture rich information about the local structural characteristics of the density map, significantly reducing the number of points to be considered, and leading to a more efficient alignment process. Meanwhile, the local feature descriptors computed based on the statistical distributions provide a comprehensive representation of the local structural variations. Using these feature descriptors, CryoAlign employs a mutual feature-matching strategy to establish correspondences between keypoints in different density maps, enabling stable alignment parameter estimation. To further refine the alignment, CryoAlign applies a point-based iterative method, aiming to bring overlapping point pairs closer together. To assess the performance of CryoAlign, comprehensive evaluations were conducted on diverse test sets, which demonstrate its high alignment accuracy for both global and local cryo-EM map alignment. In comparison to other alignment methods such as gmfit, fitmap, and VESPER, CryoAlign stands out by providing more precise superimposition of density maps while maintaining a lower failure ratio.

## Results

### Overview of the CryoAlign procedure

Figure 1a illustrates the workflow of CryoAlign. When provided with a density map, CryoAlign applies uniform sampling to generate a set of initial grid points, which act as the starting positions for the subsequent alignment process. At each sampled grid point, a corresponding density vector is assigned to reflect the trend of changes in density within its vicinity. These density vectors are derived from VESPER, which demonstrates their effectiveness as a representation of the local density variations around the grid points. However, the excessive number of initial grid points and the limited representation range of density vectors make them unsuitable for direct alignment. CryoAlign uses a mean shift algorithm[29] to identify local dense points and applies a density-based spatial clustering method[30] to find cluster centers as the key points of point clouds (see the "Methods" section). The key points extracted by CryoAlign are chosen to consider both the

distribution of density values and the connectivity of points, providing a rough representation of the protein backbones. Next, local spatial structural feature descriptors are calculated on the extracted key points by block-wise analyzing the distribution of density vectors within their vicinity. Compared to vectors, local feature descriptors capture structural information from multiple neighboring points instead of just a single grid point. This approach provides a more distinctive and comprehensive description of the local region, effectively improving the accuracy of the alignment results. Finally, CryoAlign implements a two-stage alignment approach to achieve accurate superimposition from coarse to fine. In the first stage, CryoAlign utilizes a mutual feature-matching strategy in the feature domain to establish correspondences between key points and efficiently estimate initial poses. This stage enables fast and stable alignment, laying the foundation for subsequent refinement. In the second stage, CryoAlign focuses on achieving the best possible superimposition. It considers the point-to-point correspondences between the initial grid points in the spatial domain and employs an iterative process to bring these points closer together. By iteratively adjusting the positions of overlapping points, CryoAlign continues to improve the alignment and strives for optimal alignment accuracy.

For a more illustrative explanation, a visual example of local alignment is provided on the right side in Fig. 1a. The two input cryo-EM maps represent the structures of *RNA polymerase-sigma54 holoenzyme* with promoter DNA closed complex. Notably, there is an additional transcription activator PspF intermediate present in the left map (EMD-3696, PDB ID:5nss), while it is absent in the right one (EMD-3695, PDB ID:5nsr). The top row of the visual example displays the grid-sampling point clouds of two maps, represented by dark points, along with their corresponding density vectors. The second row showcases the extracted key points, represented by colored points, and presents an example of a spatial structural feature histogram pair. These feature histogram pairs are used for alignment by filtering and selecting the most relevant and informative feature pairs. Following the direction of the hollow arrows, the two point clouds are aligned based on the filtered feature pairs. The coarse alignment stage provides an initial alignment that is approximately correct, although imperfect, with a high degree of overlap between the structures. Subsequently, the point-based stage is employed to refine the alignment and achieve the best possible superimposition by minimizing the distances between corresponding point pairs. Furthermore, for better visual evaluation, the corresponding PDB atom structures transformed by the alignment parameters are also attached in the example.

### Datasets of density maps and metrics

**Datasets.** To evaluate the performance of global and local alignment, we utilized the cryo-EM maps from the datasets provided by VESPER, which are specifically designed for global and local density map search. We began by filtering maps without fitted PDB atom models[31] and focused on collecting maps with a resolution higher than 10 Å. As a result of the filtering process, we obtained two datasets for evaluation: the global alignment dataset, which consists of 64 pairs of cryo-EM maps, and the local alignment dataset, which contains 201 map pairs. In Table 1, we present the statistical information for these map pairs. The first column, labeled "Res. range", indicates the resolution range of the input maps. The column labeled "Cross res." indicates whether the input pairs are from different resolution ranges. Using these two datasets, we assessed the performance of our alignment method indirectly by analyzing the fitted PDB models, both quantitatively and qualitatively. Furthermore, to evaluate the algorithm's performance in atomic model fitting, we also utilized intermediate-resolution protein complexes datasets provided by He[14]. We selected eight protein complexes of 4.0–8.0 Å, and each has 2–5 single chains. By leveraging these diverse datasets, we are able to comprehensively evaluate the alignment performance of our method across different scenarios and applications.

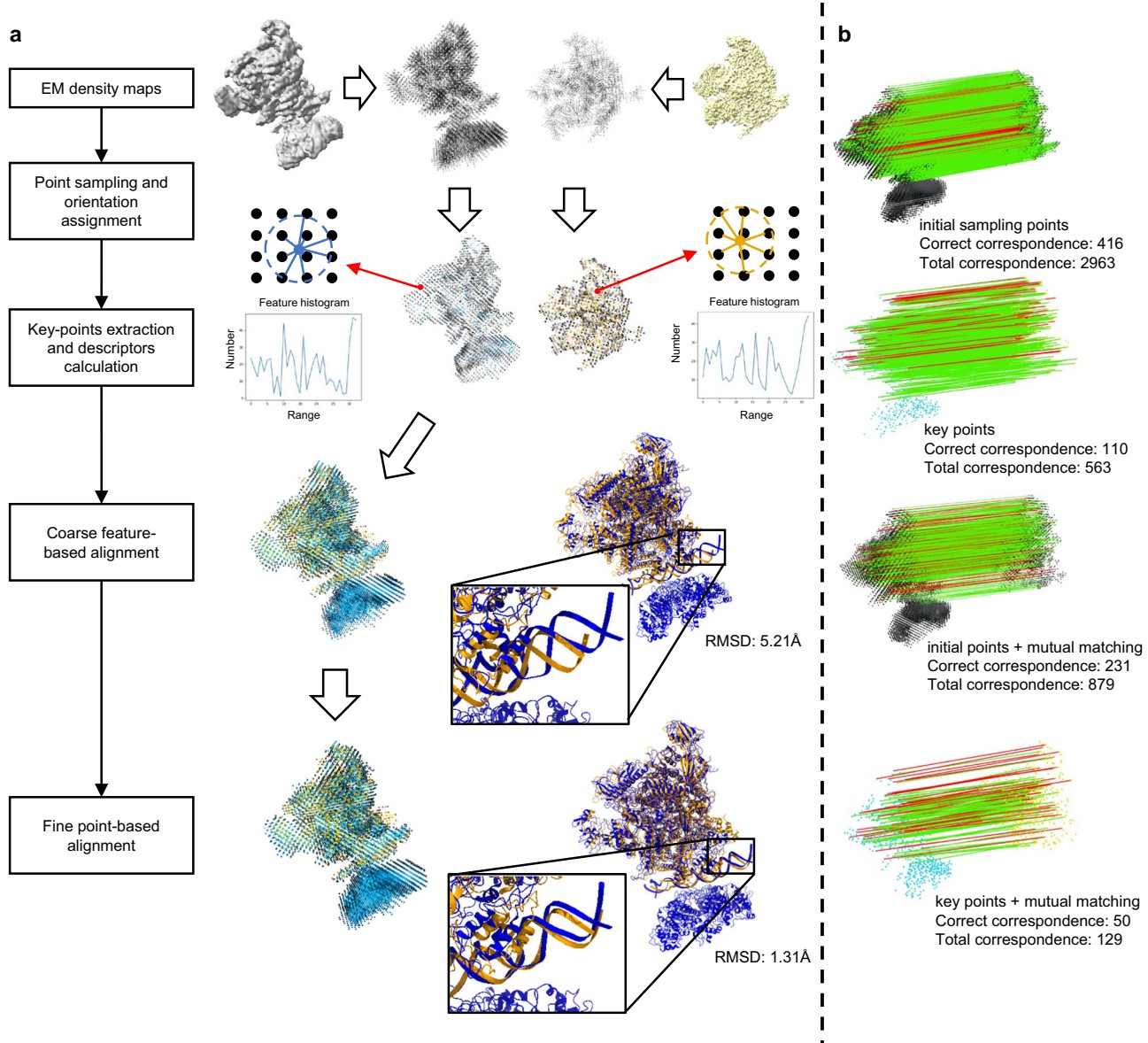

**Fig. 1 | Overview of CryoAlign. a** Flowchart of CryoAlign. A visual example of *RNA polymerase-sigma54 holoenzyme* and promoter DNA closed complex with (EMD-3696, PDB ID:5nss, left) and without (EMD-3695, PDB ID:5nsr, right) transcription activator PspF intermediate is provided on the right. The input of CryoAlign is a pair of cryo-EM density maps. First, initial point clouds are sampled by a given interval and density vectors are computed for all points. Then, clustering algorithms are applied to extract key points that represent the rough backbones of the structures. Local spatial structural feature descriptors are calculated to capture the local structures around these key points. Using the extracted feature descriptors and the mutual feature matching technique, CryoAlign robustly and efficiently computes the initial pose parameters. Finally, CryoAlign generates the best superimposition by iteratively shifting the corresponding points closer together. The alignment parameters are then applied to the fitted atom models, directly illustrating the alignment performance. **b** The proportion of correct correspondences. In the visual example, the lines between points represent the estimated correspondences, with correct correspondences labeled in red and false ones labeled in green. From top to bottom, four cases with only initial points, with only extracted key points, with a combination of initial points and mutual matching, and with a combination of key points and mutual matching are listed.

**Alignment metric.** To quantitatively evaluate the alignment performance, the ground truth for the superimposition is defined by computing the transformation parameters using MM-align[32] on fitted atom models. We then calculated the root mean square distance (RMSD) between the ground truth and the alignment results obtained by different methods. Notably, we considered an alignment as a failure if the RMSD exceeds 15 Å. This threshold helps to identify cases where the alignment deviates significantly from the ground truth. Additionally, to provide a more intuitive visualization, the fitted PDB structures were transformed using the alignment parameters, enabling a direct comparison of the aligned structures.

## Global alignment accuracy

First, we thoroughly assessed the alignment performance of CryoAlign in the pre-collected dataset. We initially sampled the density maps with an interval of 5 Å, which provides sufficient spatial distribution information for global alignment. Figure 2a shows the mean number of initial sampling points and extracted key points as the size of inputted density maps increases. After key point extraction, the point clouds typically decrease in size to around 10–20% of the initial points, making subsequent calculations more efficient. Furthermore, the distribution of key points roughly follows the structures of protein backbones, leading to more stable and accurate feature

correspondence establishment. Figure 2b presents a comparison of the feature correspondence accuracy under different scenarios, including different point cloud representations and the utilization of mutual feature matching. The orange and red curves are consistently positioned to the right of the other two curves, indicating that the utilization of key points can mitigate feature mismatches caused by excessive sampling. Additionally, the mutual feature matching strategy considers point pairs that are closest to each other in the feature domain, further enhancing the accuracy of correspondence estimation. The red curve, which represents the combination of key points

**Table 1 | Resolution statistical distribution of datasets**

| Res. range | Global alignment | Local alignment |
|---|---|---|
| <5 Å | 35 | 122 |
| 5–10 Å | 16 | 14 |
| Cross res. | 13 | 65 |

*Res. range,* resolution range. The "Cross res." means that the input pairs are from different resolution ranges. The number of density maps is counted based on the resolution range.

and mutual strategy in Fig. 2b, demonstrates that the correct matching ratio generally falls within the 20–50% range, which is acceptable for robust initial pose estimation.

CryoAlign adopts a two-stage alignment architecture to achieve precise pose estimation. The aforementioned key points based feature matching is utilized in the first stage, and provides a robust but relatively coarse pose. This stage serves as a foundation for the alignment process. In the second stage, CryoAlign shifts its focus to the initial sampling points after transformation, aiming to bring the two points sufficiently close. By combining these two stages, CryoAlign generates a more accurate superimposition of the density maps. Figure 2c collects the RMSD distributions of one-stage alignment and two-stage alignment. Almost all data points are located along or above the dashed line, illustrating that the second stage refinement consistently improves the alignment accuracy. Meanwhile, the larger bubbles are mostly concentrated in the range ≤3 Å, showing the key role of point-based correspondences in the spatial domain in high precision alignment. Moreover, thanks to the initial pose estimation provided by the first stage, the second stage of point-based alignment requires less time to converge. Figure 2d presents the distribution of execution time

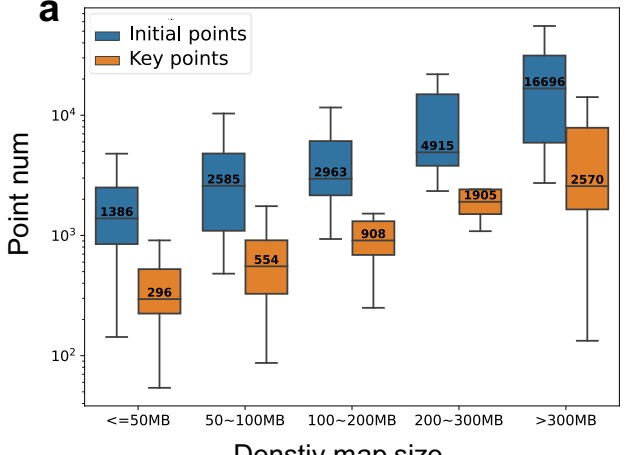

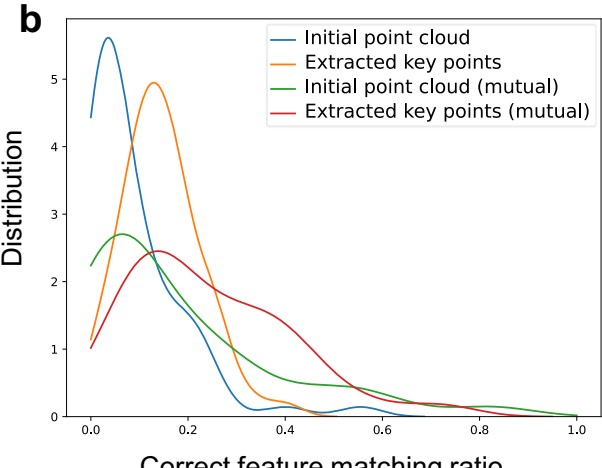

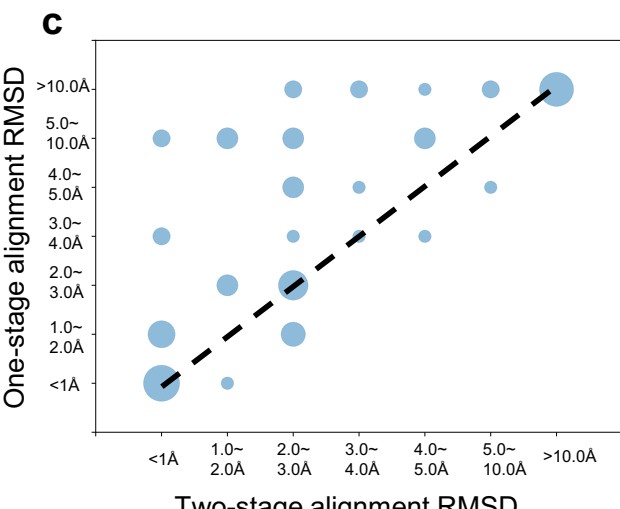

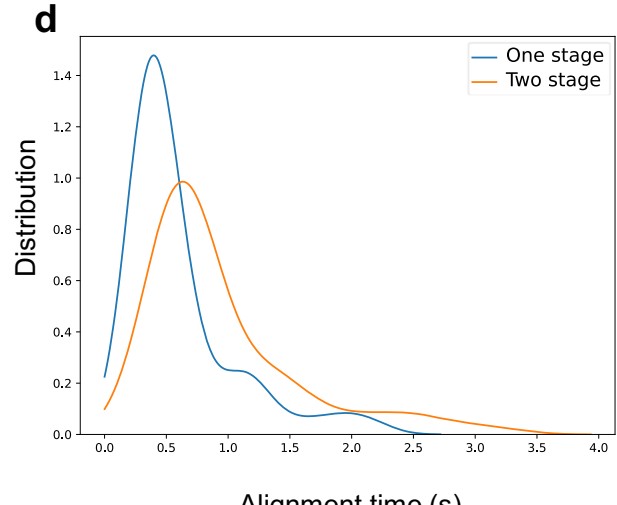

**Fig. 2 | Global alignment performance of CryoAlign. a** The number of points with the increasing density map size. Initial points (blue box) and key points (orange box). For map size groups "<=50 MB", "50–100 MB", "100–200 MB", "200–300 MB" and ">300 MB", the sample sizes N = 27, 17, 54, 80 and 15. The center, lower and upper lines in each box indicate the median, the first quartile and the third quartile, respectively. The number inside each box refers to the mean value. The whiskers show the 2.5% and 97.5% quantiles. **b** The correct ratio distribution of four different feature matching strategies. Only initial points (blue line), only key points (orange line), initial points + mutual matching (green line) and key points + mutual matching (red line). **c** Comparison of accuracy between one-stage alignment and two-stage alignment. Each data point's size corresponds to the count of combinations within specific RMSD ranges. **d** The execution time distributions of one-stage alignment and two-stage alignment.

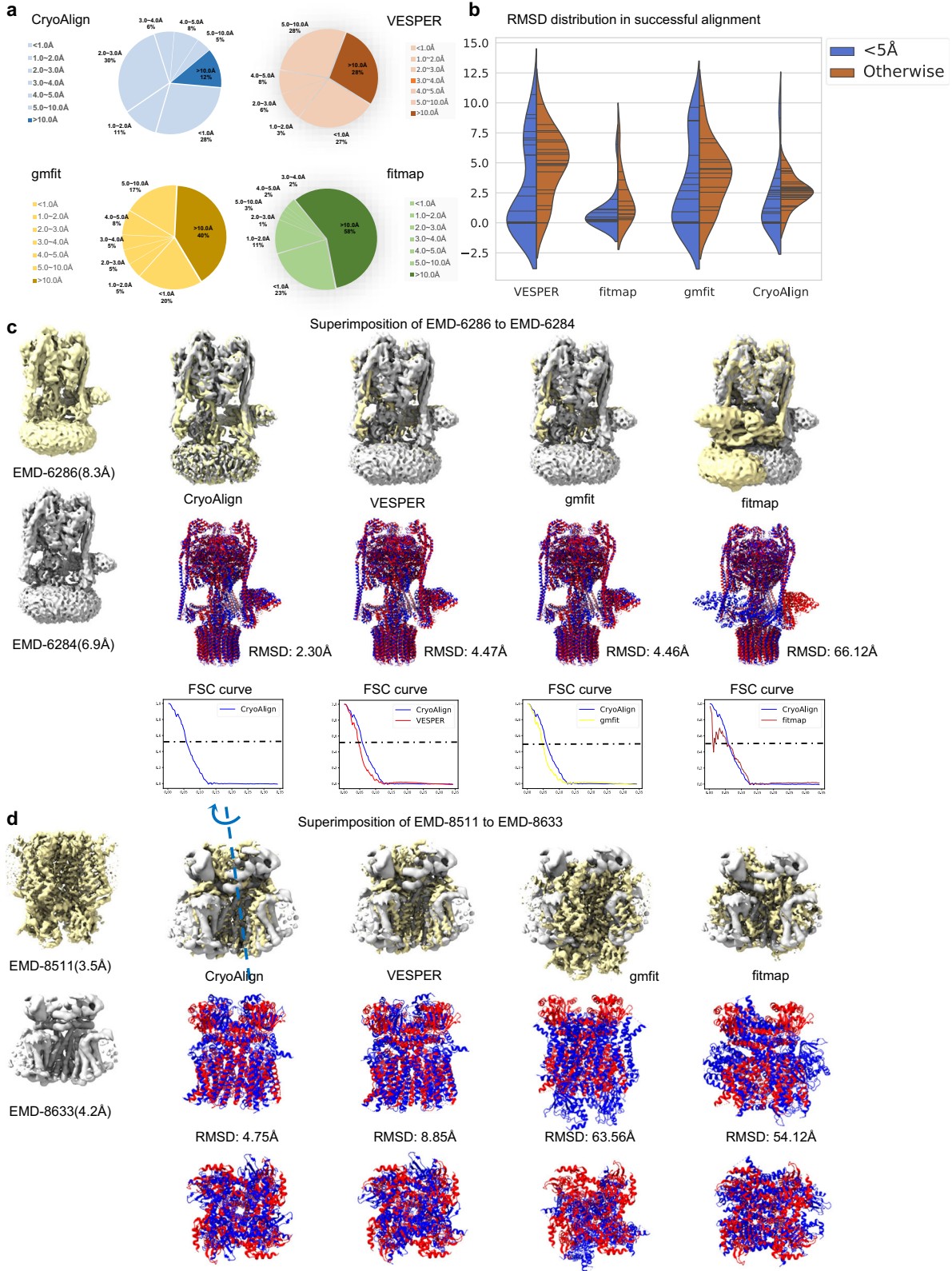

for the alignment processes, revealing that the duration of the second stage is acceptable considering the improvement in accuracy.

Figure 3a presents the RMSD distributions of CryoAlign and other comparative methods VESPER, gmfit and fitmap in global alignment datasets. For VESPER, the sampling and initial rotation intervals were set to 5 Å and 10°, respectively. gmfit was run with 20 Gaussians and parameter -maxsize 64, which are the settings in the Omokage map

web server. For fitmap, we took 20 random poses as the initial placements. The pie charts provide an overview of the alignment results for different methods, with the dark sections representing the failure proportion (RMSD larger than 10 Å) and the shallow sections representing acceptable results. The results from fitmap exhibit a highly polarized distribution, with a majority of cases falling into the >10.0 Å and <2.0 Å ranges. This indicates strong dependence on the quality of

**Fig. 3 | Global alignment performance of compared methods. a** The RMSD distribution of the compared methods CryoAlign, VESPER, gmfit and fitmap. The sectors colored dark are the failure proportion (RMSD larger than 10 Å) of the methods, with CryoAlign/VESPER/gmfit/fitmap being 12%/28%/40%/58%, respectively. Meanwhile, RMSD smaller than 3 Å can be considered as high-quality alignment, with CryoAlign/VESPER/gmfit/fitmap being 69%/36%/30%/35%, respectively. **b** The violin plot illustrates the RMSD values in successful alignment for each method, split by map resolution. Each line represents a data point. Notably, the regions below zero hold no meanings, which are merely the result of distribution estimation. **c** The left example is the density map pair for the same state of *Yeast V-ATPase* (EMD-6286, PDB ID:3j9v and EMD-6284, PDB ID:3j9t). There is little difference between the two maps. The alignment accuracy is evaluated by FSC curves on the right. The RMSD of CryoAlign/VESPER/gmfit/fitmap is 2.30/4.47/4.46/66.12 Å, respectively. **d** The right example is the density map pair for different states of *Cyclic Nucleotide-Gated Ion Channel* (EMD-8632, PDB ID:5v4s and EMD-8511, PDB ID:5u6o). Accurate rotation estimation is needed here. The RMSD of CryoAlign/VESPER/gmfit/fitmap is 4.75/8.85/63.56/54.12 Å, respectively.

the initial poses provided. gmfit shows a relatively average distribution, but it has the smallest section in the <1.0 Å range, suggesting lower accuracy due to its blurred Gaussian representation. Compared to gmfit and fitmap, VESPER exhibits a significant improvement in the success rate, reducing the failure proportion to 28%. However, its grid-sampling interval (5 Å) leads to RMSD values primarily falling within the 3.0–10.0 Å range. In contrast, CryoAlign achieves the lowest failure ratio and highest accuracy, with the majority of RMSD values concentrated below 3.0 Å. The violin plot in Fig. 3b displays the fitted distributions of RMSD values in successful alignment for the compared methods. The blue segments reflect the RMSD values of input maps, both having resolution higher than 5 Å. The brown segments represent the remaining scenarios. Notably, all methods exhibit lower RMSD values in blue areas than in brown areas, consistent with the expectation that higher-resolution maps yield more reliable density values. However, compared to fitmap, both VESPER and gmfit tend to generate more results with RMSD values in the range above 6 Å in blue areas. This suggests their limited ability to exploit the advantage offered by high resolution, primarily due to their neglect of local structural characteristics. gmfit, for example, relies on merely 20 Gaussians to fit the overall shape of density maps, ignoring detailed local structures. Despite the utilization of density vectors in VESPER, the summation operation significantly dilutes the influence of small structures. In contrast, CryoAlign aims to capture detailed structural information using local spatial feature descriptors. Higher-resolution maps bring clearer structures, making the corresponding feature description more distinctive. As shown in Fig. 3b, the blue area of CryoAlign is primarily below 3 Å, showing the powerful ability to capture detailed structural information.

Table 2 summarizes the average RMSD for different resolution ranges and execution times of CryoAlign and the compared methods. The first column of the table represents the resolution range of the two input density maps, and "Cross resolution" indicates that the maps have different resolution ranges. Notably, we define an RMSD larger than 10 Å as an alignment failure. Among the methods evaluated, fitmap has the lowest average RMSD in the first two rows, but it also has a failure rate of ~50%. This is because fitmap relies heavily on the given

initial poses, which are based on the domain knowledge of researchers. Without accurate initial poses, fitmap tends to produce poor alignment results. VESPER effectively reduces the failure ratio by exploring a large number of candidate alignment parameters. However, the 5 Å grid sampling significantly constrains the upper limits of its alignment accuracy. CryoAlign achieves the second-lowest average RMSD after fitmap, demonstrating the ability of feature descriptors to overcome the limitations derived from sampling intervals. The clustering process detects the key backbone positions as anchor points, forming the solid foundation for sub-voxel accuracy estimation. The subsequent parameter estimation based on correct point correspondences ensures the implementation of this precision. Regarding the failure ratio, the improvement of CryoAlign over VESPER is mainly attributed to its utilization of density vectors. VESPER collects vectors located on the same grid points to measure similarity, eliminating the influences of non-overlapping regions. Combined with a rough rotation estimation, this process makes VESPER easily neglect the small structures, which could be the key to distinguishing the difference between similar chains. In contrast, CryoAlign utilizes information from nearly all points by collecting neighboring points in local spatial feature construction. Furthermore, establishing correspondences in the feature domain forces CryoAlign to focus on the points with unique or distinguished descriptions. These key structures help the algorithm locate the correct superimposition. We also collected execution time information for the point generation process and the alignment stage of the four methods. gmfit models the density maps via combinations of multiple Gaussian kernels, which provide a rough representation of the 3D shape. Due to a relatively small number of weights and parameters used, it executes the fastest but with lower accuracy. The execution time of fitmap depends mainly on the number of initial poses, and in our experiments, using 20 initial poses strikes a balance between accuracy and efficiency. Compared to other methods, CryoAlign takes considerable time in point extraction due to the additional key point descriptor computation. However, in the alignment stage, CryoAlign executes much faster. This is mainly because CryoAlign directly estimates the transformation parameters based on point correspondences, while VESPER needs to scan the entire translation/rotation spaces. In summary, CryoAlign outperforms the compared methods in terms of both accuracy and efficiency in global alignment, with comprehensive consideration of both accuracy and efficiency.

### Examples of global alignment

For a direct and fair comparison, we collected test examples of different resolutions in VESPER (Table 2 in its manuscript). Table 3 summarizes the RMSD of the best superimposition achieved by CryoAlign compared to VESPER, gmfit, and fitmap. The parameter combination used for VESPER was set to (1 Å, 10°), and the performances of gmfit and fitmap were directly taken from the recommendations from their paper. In cases where the input maps have the same resolution range (either <5 Å or 5–10 Å), CryoAlign achieves results that are closest to the ground truth superimposition. Even when the given maps have different resolutions, CryoAlign still provides acceptable pose estimation. This comparison demonstrates the effectiveness of CryoAlign in achieving accurate and reliable alignment results, especially when

### Table 2 | Alignment evaluation in global dataset

| Res. range | CryoAlign(Å)/ failure | VESPER(Å)/ failure | gmfit(Å)/ failure | fitmap(Å)/ failure |
|---|---|---|---|---|
| <5 Å | 1.69/18.4% | 2.853/25.71% | 3.01/37.14% | **0.78**/48.57% |
| 5.0–10.0 Å | 2.88/**6.25%** | 5.09/25% | 7.59/25% | **0.82**/50% |
| Cross res. | **2.23/0%** | 4.53/23.08% | 3.58/46.15% | 3.9/61.54% |
| Time | CryoAlign(s) | VESPER(s) | gmfit(s) | fitmap(s) |
| Extract points | 18.9 | 3.1 | 5.35 | - |
| Alignment | 0.94 | 202.5 | 0.213 | 60.12 |
| Total time | 19.84 | 205.6 | 5.56 | 60.12 |

There are two metrics calculated in the alignment evaluation, average RMSD and failure ratio. For RMSD, the smaller value means better alignment accuracy; for the failure ratio, the smaller value indicates higher stability. In the tables of this manuscript, for better presentation, the best results are marked in bold and the second best ones are underlined.

**Table 3 | Examples of global map alignment**

| Res. range | Map 1 IDs | Map 2 IDs | PDB RMSD(Å) | RMSD(Å) | | | |
|---|---|---|---|---|---|---|---|
| | | | | CryoAlign | VESPER(1 Å) | gmfit | fitmap |
| <5 Å | 3240/5fn5 | 2677/5a63 | 1.91 | **1.53** | 2.21 | 2.63 | 2.9 |
| | 8881/5wpq | 8764/5w3s | 2.08 | **0.68** | 1.12 | 1.19 | 56.99 |
| | 9515/5gjw | 6475/3jbr | 4.37 | **0.72** | 2.31 | 2.95 | 97.48 |
| 5–10 Å | 8744/5vy8 | 8267/5kne | 3.44 | **0.39** | 0.86 | 2.3 | 73.67 |
| | 6284/3j9t | 8724/5vox | 5.13 | 1.06 | 2.79 | 5.05 | **1.04** |
| | 3342/5fwm | 3341/5fwl | 1.45 | **0.51** | 0.56 | 3.6 | 4.98 |
| Cross res. | 8784/5w9i(3.6) | 8789/5w9n(5.0) | 8.33 | **0.05** | 2.84 | 4.69 | 79.51 |
| | 9515/5gjw(3.9) | 6476/3jbr(6.1) | 4.37 | 4.31 | **3.12** | 6.06 | 64.19 |
| | 3238/5fn3(4.1) | 2678/5a63(5.4) | 0.68 | **2.73** | 3.34 | 3.22 | 3.68 |

dealing with maps of the same resolution range. Meanwhile, it showcases the robustness of CryoAlign in handling different resolutions and its ability to estimate accurate poses even in challenging scenarios.

Furthermore, Fig. 3c, d shows two classic examples of global alignment. The first involves a density map pair representing the same state of *Yeast V-ATPase* (EMD-6286, PDB ID:3j9v and EMD-6284, PDB ID:3j9t). These maps are nearly identical, with only minor differences caused by molecular dynamics or imaging variations. In this case, the accuracy of translation parameter estimation plays a crucial role in alignment accuracy. Both CryoAlign and VESPER show excellent visual performances in terms of superimposition. However, the difference in RMSD is reflected mainly in the Fourier Shell Correlation (FSC) curve. The FSC figure below the example illustrates that the blue curve, representing CryoAlign, is consistently positioned to the right of the red curve, indicating more accurate alignment parameters. This is because the grid sampling interval (5 Å) limits the upper bound of translation estimation in VESPER, while CryoAlign gets rid of it by estimating parameters in the feature domain. The second example involves a density map pair representing different states of the *Cyclic Nucleotide-Gated Ion Channel* (EMD-8632, PDB ID:5v4s and EMD-8511, PDB ID:5u6o). These maps exhibit structural similarities but have significant contour differences. Additionally, there is rotational invariance around an axis, which imposes higher requirements on rotation parameter estimation. For comparison, we provide two different viewing directions of the PDB atom model superimposition. The left view represents the ordinary viewing direction, while the right view represents the rotation axis view. From the ordinary viewing direction, both CryoAlign and VESPER demonstrate accurate translation parameter estimation. However, from the rotation axis viewing direction, VESPER exhibits a larger RMSD, indicating poor rotation parameter estimation. One possible reason for this discrepancy is that the fixed rotation interval of VESPER may constrain the fine rotation estimation. Meanwhile, density vectors, reflecting the trend of density around merely a small area, cannot provide sufficient discrimination on the overall rotation. In contrast, CryoAlign utilizes the orientation distribution of local regions as features, allowing for a more accurate estimation of rotation parameters.

**Local alignment accuracy**

Regarding the local alignment, it is important to consider the size difference between the input density maps. If the size of the smaller map occupies more than 40% of the size of the larger map (volume ratio), the accuracy of feature matching remains similar to that of global alignment in most cases. However, if the size difference is too large, it becomes challenging for feature-based alignment to find an acceptable superimposition in a single attempt. Figure 4a illustrates the higher failure probability as the volume ratio decreases. This is because the candidate feature descriptors from the larger map can easily interfere with the smaller number of feature queries. To address accurate local alignment,

CryoAlign treats it as a global retrieval problem within a small "dataset". It adopts a translational mask as a simple segmentation scheme for the larger point cloud, as shown in Fig. 4c. The two-stage alignment process is then used to calculate a series of pose parameters. Based on this collection of parameters, CryoAlign measures the similarity scores across all superimpositions and selects the top one as the output. Moreover, Fig. 4b demonstrates the masking strategy not only helps to find the best superimposition in cases with low volume ratios but also improves the alignment accuracy in cases with high volume ratios. This discovery suggests the presence of numerous mismatches in feature matching, even within the context of global alignment, of which a discussion is made in "Exploration of local spatial features". Similar to global alignment, the violin plots of successful alignment's RMSD values for compared methods are demonstrated in Fig. 4d. VESPER exhibits highly-close-shaped blue and brown areas, suggesting the exhaustive search methods are not sensitive to resolution, in which the predefined rotation/translation intervals limit the exploration of high-resolution information. fitmap shows lower accuracy than global alignment as voxel-based cross-correlation can be easily affected by neighboring voxels, especially in small volume ratio situations. In the case of CryoAlign, the majority of its brown areas are close to the blue ones, which means similar accuracy in different resolution, and indicates that the masking strategy in CryoAlign, to some extent, compensates for the impact of relatively low resolution.

The average RMSD and failure information for local alignment are presented in Table 4. In comparison to global alignment, both gmfit and fitmap exhibit high failure ratios, ranging from 80% to even 100%. This highlights the difficulty of directly aligning two density maps in local alignment. Non-overlapping regions significantly affect the correlation calculation and further destroy the correspondence establishment. In contrast, VESPER employs a similarity measurement based on matched vectors in overlapping regions to eliminate that interference, enabling its applicability in the local alignment. Similarly, CryoAlign generates a series of candidate parameters using a translational mask and selects the best one. This straightforward segmentation strategy effectively transforms the local alignment problem into multiple global alignment problems, ensuring the accuracy of the feature-matching stage to a certain extent. Notably, the feature construction based on neighboring points is inevitably influenced by points beyond overlapping regions, especially when the smaller volume is entirely embedded within the larger one. Fortunately, the extracted key points are mostly located in the internal regions of point clouds due to clustering processes. This ensures the predominance of useful points in the vicinity and prevents the failure of feature matching. Similar to global alignment, CryoAlign demonstrates lower average RMSD values, indicating superior performance compared to VESPER within the same sampling interval.

Two examples of local alignment are shown in Fig. 4e, f. In the first example, we aim to superimpose the Vo region of the *V-ATPase* (EMD-

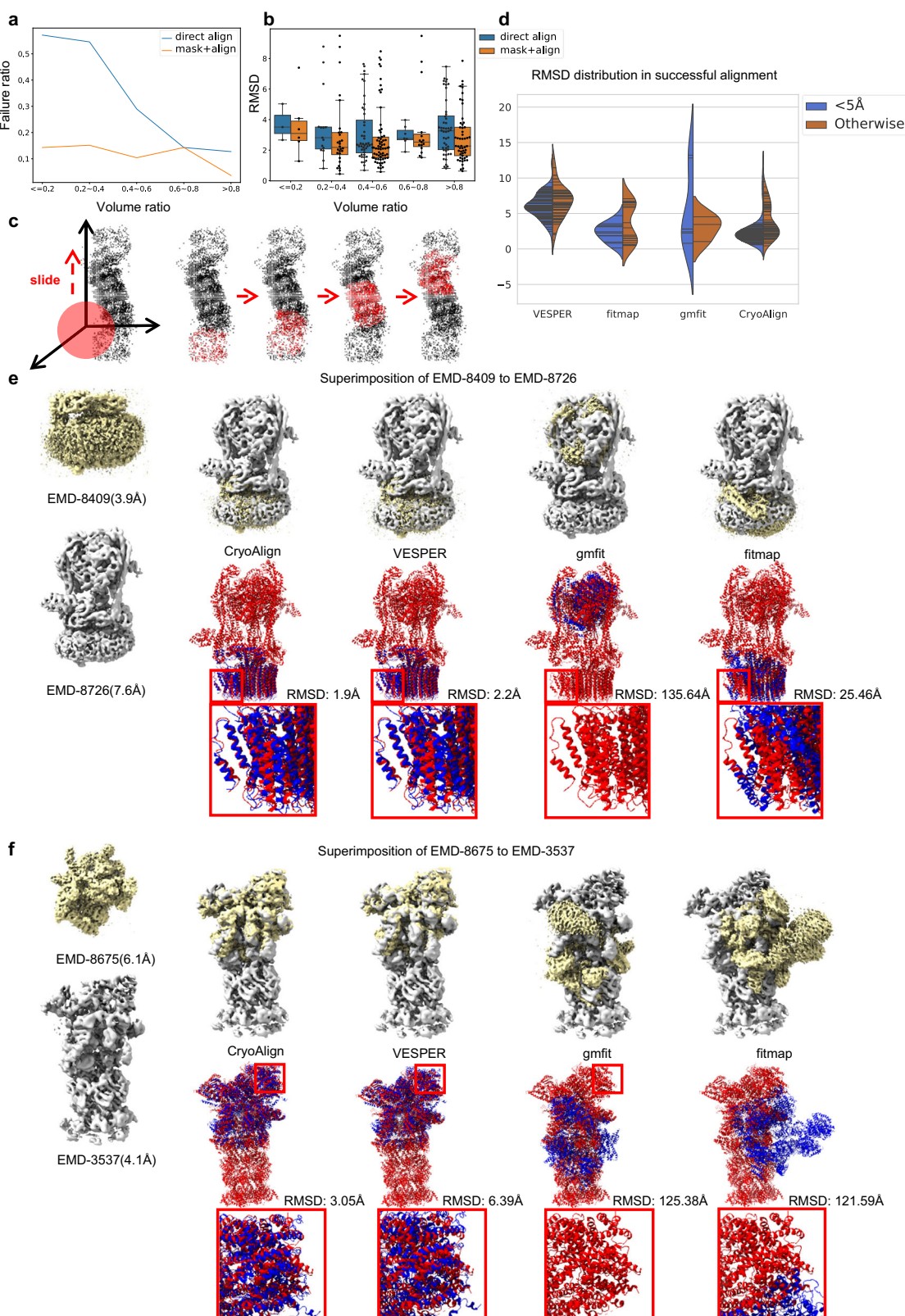

8409, PDB ID:5tj5) onto the complete *V-ATPase* (EMD-8726, PDB ID:5voz). Despite EMD-8409 occupying less than 40% of the volume of EMD-8726, its distinct fence-like 3D structure makes it stand out within the complete V-ATPase map. Both CryoAlign and VESPER achieve high alignment accuracy, with RMSD values of ~2 Å, significantly lower than the sampling interval of 5 Å. gmfit fails to capture the local structures by using merely 20 Gaussians, and completely misplaces the source

map. fitmap, despite accepting an approximate initial pose, also fails due to excessive focus on the overlapping region. The changes of density depict the local structures better than voxel values. Upon observing the enlarged PDB models, we can see that fitmap attempts to align the right side better while neglecting the left side. The second example involves the alignment of the *26S proteasome* regulatory particle (EMD-8675, PDB ID:5vhh) and the *26S proteasome* of

**Fig. 4 | Local alignment. a** The relation between failure probability and volume proportion of the smaller map to the larger one. The blue curve is direct alignment without cutting. The orange curve is multiple alignment with a translational mask. **b** Comparison of alignment accuracy between two alignment strategies, direct alignment and multiple alignment with mask. For volume ratio groups "<=0.2", "0.2–0.4", "0.4–0.6", "0.6–0.8", and ">0.8", the sample sizes $N$ = 7, 33, 77, 29, and 55. The center, lower and upper lines in each box indicate the median, the first quartile and the third quartile, respectively. The number inside each box refers to the mean value. The whiskers show the 2.5% and 97.5% quantiles and each black dot represents a data point. **c** Sketch of the translational mask. The mask moves in a given interval along the axis and part of the larger point cloud is taken for alignment. The extracted points are labeled in red and the remaining ones are black. **d** The violin plot illustrates RMSD values in successful alignment for each method, split by map resolution. Each line represents a data point. Notably, the regions below zero hold no meanings, which are merely the result of distribution estimation. **e** The first example is superimposing the Vo region of the *V-ATPase* (EMD-8409, PDB ID:5tj5) on the complete *V-ATPase* (EMD-8726, PDB ID:5voz). Although the volume ratio is smaller than 50%, the distinct fence-like 3D structure makes EMD-8409 distinctive from EMD-8726. The RMSD of CryoAlign/VESPER/gmfit/fitmap is 1.9/2.2/135.64/25.46 Å, respectively. **f** The second example is to align *26S proteasome* regulatory particle (EMD-8675, PDB ID:5vhh) and *26S proteasome* of Saccharomyces cerevisiae in the presence of BeFx (EMD-3537, PDB ID:5mpc). The volume ratio is ~50%, but it is still difficult to align them using traditional methods. The RMSD of CryoAlign/VESPER/gmfit/fitmap is 3.05/6.39/125.38/121.59 Å, respectively.

Saccharomyces cerevisiae in the presence of BeFx (EMD-3537, PDB ID:5mpc). The failures of gmfit and fitmap demonstrate that when the smaller map occupies approximately or less than 50% of the larger one, it becomes challenging for conventional methods to correctly align them. VESPER tries to eliminate interference from non-overlapping regions by scanning the entire rotation/translation space, but the fixed translation and rotation intervals limit its precision. In contrast, CryoAlign employs a correspondence-based method to estimate "sub-voxel" transformation parameters, resulting in a lower RMSD.

## Application in map comparison

Accurate alignment of density maps is an essential step in heterogeneity analysis or 3D classification. Existing software often employs cross-correlation-based methods to directly quantify voxel differences between maps. This approach typically works well when the maps are roughly pre-aligned or the differences are not sufficiently significant. In fact, cross-correlation methods still encounter issues arising from inadequate initial poses. As a point cloud based approach, CryoAlign might not provide the same level of precise superimposition as cross-correlation methods, due to information loss resulting from the point sampling process. However, CryoAlign has the ability to achieve a sufficiently close map superimposition, which could potentially serve as an initial pose for subsequent refinement processes.

In Fig. 5, we present examples showing different states of *bL17-limited ribosome assembly* intermediates[33]. Figure 5a presents a comparison between state #16 (EMD-24492) and state #20 (EMD24491). These two states are quite similar, with the primary distinction being an area in the upper right corner. We computed the difference maps for both scenarios: source map−target map and target map−source map. The differences were defined as changes in molecular weight, which directly correspond to the voxel-based difference densities and are calculated using 0.81 Da/Å³. Notably, CryoAlign achieves a comparable superimposition to fitmap, while VESPER produces a less accurate result. In Fig. 5b, we analyze the comparison between state #1 (EMD-24671) and state #28 (EMD-24561). Substantial differences exist between the two maps, posing a challenge for cross-correlation-based methods such as fitmap. Compared to VESPER, CroAlign offers a better superimposition, which can serve as an acceptable initial position for the subsequent refinement. In the "Difference map 2" column, the molecular weight of CryoAlign is significantly lower than the weight of VESPER. Furthermore, the combination of CryoAlign and fitmap yields the lowest weight, demonstrating the feasibility of integrating these two methods.

Additionally, the high-precision alignment of CryoAlign enables an accurate map comparison of compared maps and helps the user more easily locate the variable regions and further analyze the conformation change. We collected a dataset of in total 42 different states of *bL17-limited ribosome assembly* intermediates from EMPAIR 10841. The 3D variance map was computed by fixing EMD-24491 as the reference map and aligning the remaining 41 conformations to it. Some examples of different states are presented in the top row of Fig. 5c. Notably, fitmap occasionally encounters alignment failures, as illustrated in Fig. 5b. Consequently, the resulting variance images exhibit a uniform numerical distribution lacking differentiation, impeding the observation of conformational changes. VESPER delivers alignment results, albeit with less accuracy, facilitating the rough identification of variable regions with higher variances in the range [20, 30]. For example, the discernible changes in the upper parts of maps are apparent through analysis of the variance image in the y-z plane. However, the relatively lower alignment accuracy of VESPER introduces potential confusion between variable and stable areas, as variances in some stable regions also fall within the range [15, 20]. In contrast, the variance slice generated by CryoAlign reveals more pronounced distinctions between variable and stable regions. Here, larger variance values are concentrated in the range [20, 35], while smaller ones predominate in the range [0, 10]. These distinguished variance differences are the key to locating the conformational changes and moving regions.

## Application in atomic model fitting

Local alignment plays a crucial role in the assembly of single chains in protein complex atom modeling. To facilitate this process, we gathered a set of density maps representing protein complexes along with their associated PDB entries. From each fitted PDB atom model, we extracted all single chains present. For every single chain, we simulated a corresponding density map using the "molmap" command in Chimera, ensuring that the resolution matches that of the target complete map. To achieve higher alignment accuracy, we set the initial sampling interval to 3 Å for both CryoAlign and VESPER. This choice is motivated by the small size of the single protein chains, where a smaller sampling interval can provide more detailed structural information.

We present two representative examples of atomic model fitting using CryoAlign. The first example involves the *pentameric ZntB transporter* (EMD-3605, PDB ID:5n9y), which consists of five single chains labeled A to E (Fig. 6a). Due to the structural similarity among the five chains, they exhibit a certain degree of rotation invariance. To account for this invariance, we provide the top five scoring parameters and indicate the rank of the best superimposition. In Fig. 6a, the rank is denoted by "(#2)" next to the RMSD value in red. The unselected top-ranked alignment results are attached in the Supplementary Material section "Ranked results in atomic model fitting". If no ranking information is given, the RMSD was calculated based on the top-scoring alignment (i.e., by default, the RMSD of the first ranking alignment was calculated). In this example, gmfit and fitmap generally fail to produce satisfactory results, highlighting the challenges of correlation construction between maps with significant volume differences. Although VESPER finds acceptable alignment parameters, the rankings of three chain results A, B and D are low. This is primarily due to the given rotation interval, which is set to 10 degrees for efficiency in the parameter searching. When the candidate chains exhibit structural similarity in rotation, the less accurate alignment provided by VESPER fails to capture the detailed structural differences by measuring the directions of matched vectors. Consequently, this leads to a lack of discrimination among the top candidate alignments. CryoAlign, on the other hand, establishes the point correspondences in the feature domain. The high-quality feature descriptors ensure the consistency and accuracy of

**Table 4 | Alignment evaluation on the local dataset**

| Res. range | CryoAlign(Å)/ failure | VESPER(Å)/ failure | gmfit(Å)/ failure | fitmap(Å)/ failure |
|---|---|---|---|---|
| <5 Å | <u>3.77</u>/9.02% | 6.07/**0.0%** | 8.05/92.6% | **2.34**/91.8% |
| 5.0–10.0 Å | **3.24/0.0%** | <u>6.48</u>/14.29% | 12.62/57.1% | 6.55/85.7% |
| Cross res. | 4.92/**12.31%** | 7.02/<u>13.8%</u> | –/100% | **4.15**/75.4% |

feature matching, enabling CryoAlign to estimate the same parameters across different masking regions. This helps CryoAlign effectively distinguish the best superimposition among candidate alignments compared to VESPER. The second example (Fig. 6b) involves the *kinase domain-like (MLKL)* protein (EMD-0868, PDB ID:6lba). It should be noted that if no ranking is provided alongside the RMSD, none of the top five scoring parameters yielded successful alignments. For instance, the second and third rows of VESPER in the example demonstrate its inability to find the correct position due to the rotational invariance. Using the same sampling interval, CryoAlign achieves more accurate alignment performance in terms of RMSD compared to VESPER. Additionally, we provide more atomic model fitting results in the Supplementary Material section "More atomic model fitting results" to demonstrate the superior alignment accuracy of CryoAlign.

Moreover, through accurate rigid transformations, multiple chains are all placed into appropriate positions, serving as the initial assembling model. This well-assembled initial model is a crucial foundation for subsequent flexible fitting, an indispensable step in high-precision atomic modeling. CroAlign can conveniently integrate with existing point cloud-based approaches[34–36] to address this requirement. A protein structure typically consists of multiple chains. First, in CryoAlign, each chain is transformed into a point cloud, and aligned to the fixed map. CryoAlign transforms these point clouds representing chains respectively and merges them into a comprehensive and larger point cloud. The assembly of point clouds is an initial model representation of the protein structure. Then, the integrated point cloud as a whole, is compared to the fixed reference to estimate displacements for each point. Finally, the motion of point clouds can be coherently translated into the atomic coordinates, as both point clouds and atoms share the same coordinate system. Interested researchers can refer to the Supplementary Material section "Extended results in flexible fitting" for the visual examples.

## Discussion

In this study, we introduced CryoAlign, a highly accurate method for aligning cryo-EM density maps at both global and local levels. CryoAlign operates by transforming the input maps into 3D points and leveraging local spatial structural feature descriptors to capture the underlying structural information effectively. The alignment process in CryoAlign is conducted in two stages. In the first stage, CryoAlign employs clustering-based key point extraction and mutual feature matching techniques to establish correspondences between the extracted key points in the feature domain. This enables CryoAlign to set a solid foundation for achieving fast and robust superimposition. In the second stage, CryoAlign focuses on establishing correct point-to-point correspondences between the sampled points in the spatial domain. By carefully building these correspondences, CryoAlign calculates the final transformation parameters, resulting in a highly precise superimposition.

CryoAlign surpasses existing methods in terms of alignment accuracy for global alignment tasks, while maintaining a good execution time. By achieving more precise density map superimposition, CryoAlign enables researchers to identify and analyze differences or changes between two maps, leading to a better understanding of biological structures. While the parameter settings used in the experiment results demonstrate the superior alignment performance of CryoAlign, it is worth noting that these settings are not necessarily optimized for all tasks or imaging environments. Users have the flexibility to explore different parameter configurations based on their specific requirements ("Parameter settings" in Supplementary Material). In addition to alignment accuracy, CryoAlign offers a scoring function that measures the similarity between two maps. This scoring function can be used in map retrieval tasks, allowing researchers to search for maps with similar characteristics or features.

For local alignment, CryoAlign employs local spatial structural feature descriptor-based alignment combined with a segmentation approach. The simple segmentation strategy using translational masks has demonstrated its effectiveness in experiments, but it may suffer from redundancy. By incorporating domain knowledge and developing a more advanced segmentation scheme, CryoAlign can achieve faster and more accurate results in local alignment tasks. Local map alignment plays a crucial role in the subunit assembly of protein macromolecular atom modeling. Since identical single chains may exist in the structure, CryoAlign provides multiple transformation candidates ranked by similarity scores. Users can evaluate each alternative superimposition and select the most suitable one based on their domain knowledge and expertise.

CryoAlign is designed to assist in further comparing, mining and modeling of the reconstructed cryo-EM density maps. Extracting valid spatial structures relies on informative density values and corresponding contour levels. Extremely low signal-to-noise ratios may make CryoAlign unable to distinguish structural information. Thus, CryoAlign cannot be applied to tasks such as sub-volume alignment in subtomogram averaging, which have been affected by extremely high noise and the "missing wedge" effect. Fortunately, the cryo-EM maps in EMDB usually have a relatively high SNR, and real-world experiments show that CryoAlign is accurate enough to handle the general cryo-EM map alignment tasks and robust to the initial orientation choice of the 3D maps and cross-resolution comparison.

In conclusion, CryoAlign offers a robust and accurate alignment solution for cryo-EM density maps with a resolution higher than 10 Å. Its capabilities in both global and local alignment make it a valuable tool for studying and analyzing structural biology cryo-EM maps. CryoAlign's ability to accurately superimpose maps enables researchers to gain deeper insights into the structural details and variations present in the maps.

## Methods
### Point cloud generation
CryoAlign starts by converting the input density map into a point cloud through uniform sampling, assigning density vectors using the mean shift equation. It then identifies key points within the point cloud using clustering techniques and computes local spatial structural feature descriptors. These key points and feature descriptors are utilized in the subsequent alignment stages to achieve accurate alignment.

**Initial density-based point generation.** The successful application of VESPER demonstrates the intensive unit vectors have the ability to capture the local structures of density maps. CryoAlign regards the uniformly sampled grid points as the point cloud and calculates unit vectors as the "density vectors" for these points. The unit vector is computed for each grid point $x_i(i=1,...,N)$ with a density value that no less than author-recommended contour level. The direction $\frac{\overrightarrow{y_i - x_i}}{|y_i - x_i|}$ of unit vector reflects the trend of density values around the grid point $x_i$, of which the $y_i$ is calculated by the following formula:

$$y_i = \frac{\sum_{n=1}^{N} k(x_i - x_n)\Phi(x_n)x_n}{\sum_{n'=1}^{N} k(x_i - x_{n'})\Phi(x_{n'})}, \tag{1}$$

where $k(p)$ is a Gaussian kernel function and $\Phi(x_i)$ is the density value of the grid point $x_i$. The $k(p)$ adjusts the influence of neighboring

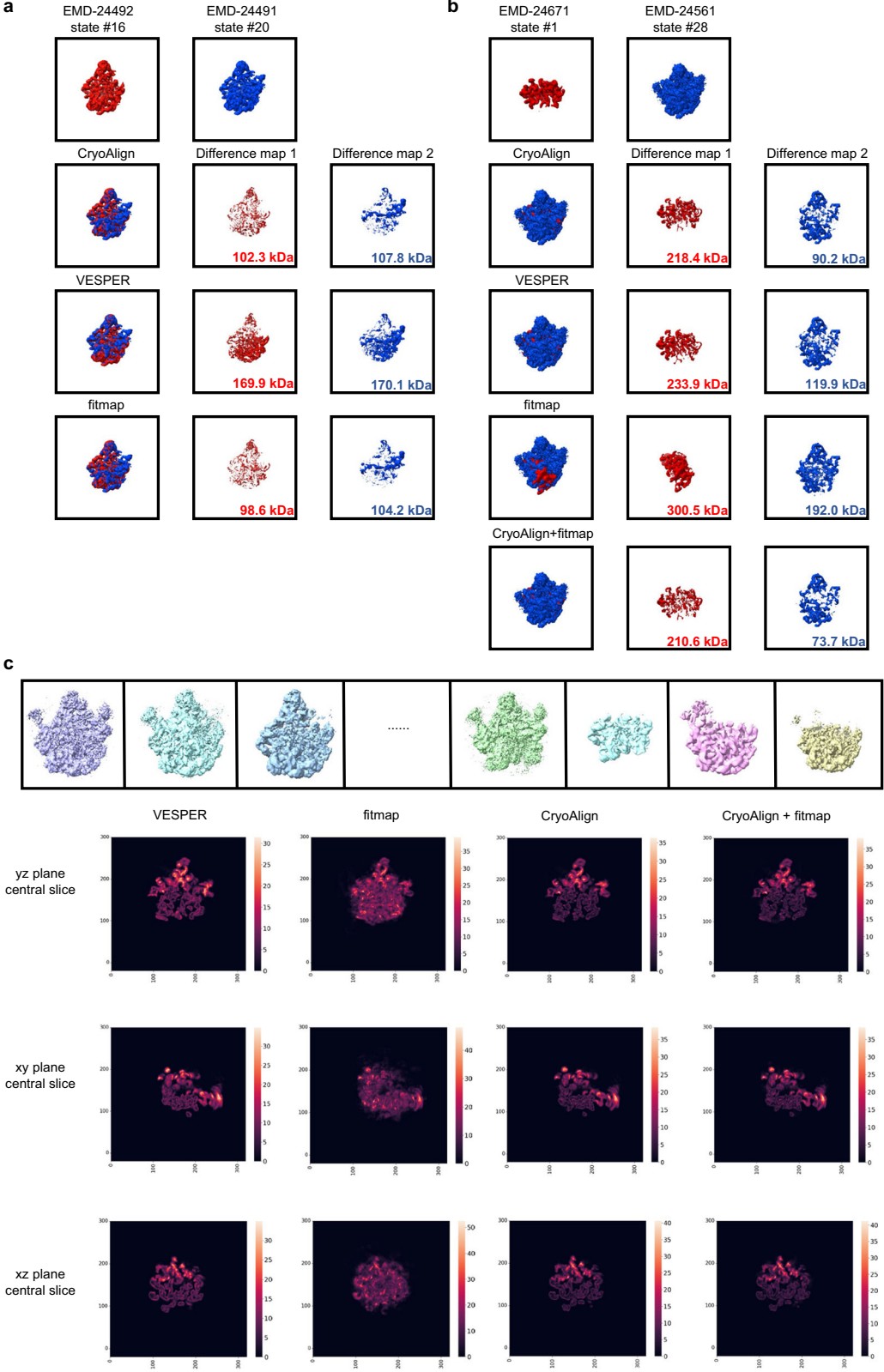

**Fig. 5 | Examples for map comparison. a**, **b** The difference map is calculated for both scenarios: source map−target map and target map−source map. The molecular weights are computed to quantify the difference. **c** 3D variance map of 42 different states of *bL17-limited ribosome assembly* intermediates. Some representative ribosome assembly intermediates of different states are selected in the top row. The 3D variance map is displayed in the central slice of the yz plane, xy plane, and xz plane for visualization. The color intensities correspond to the variance values, with brighter colors indicating higher variances.

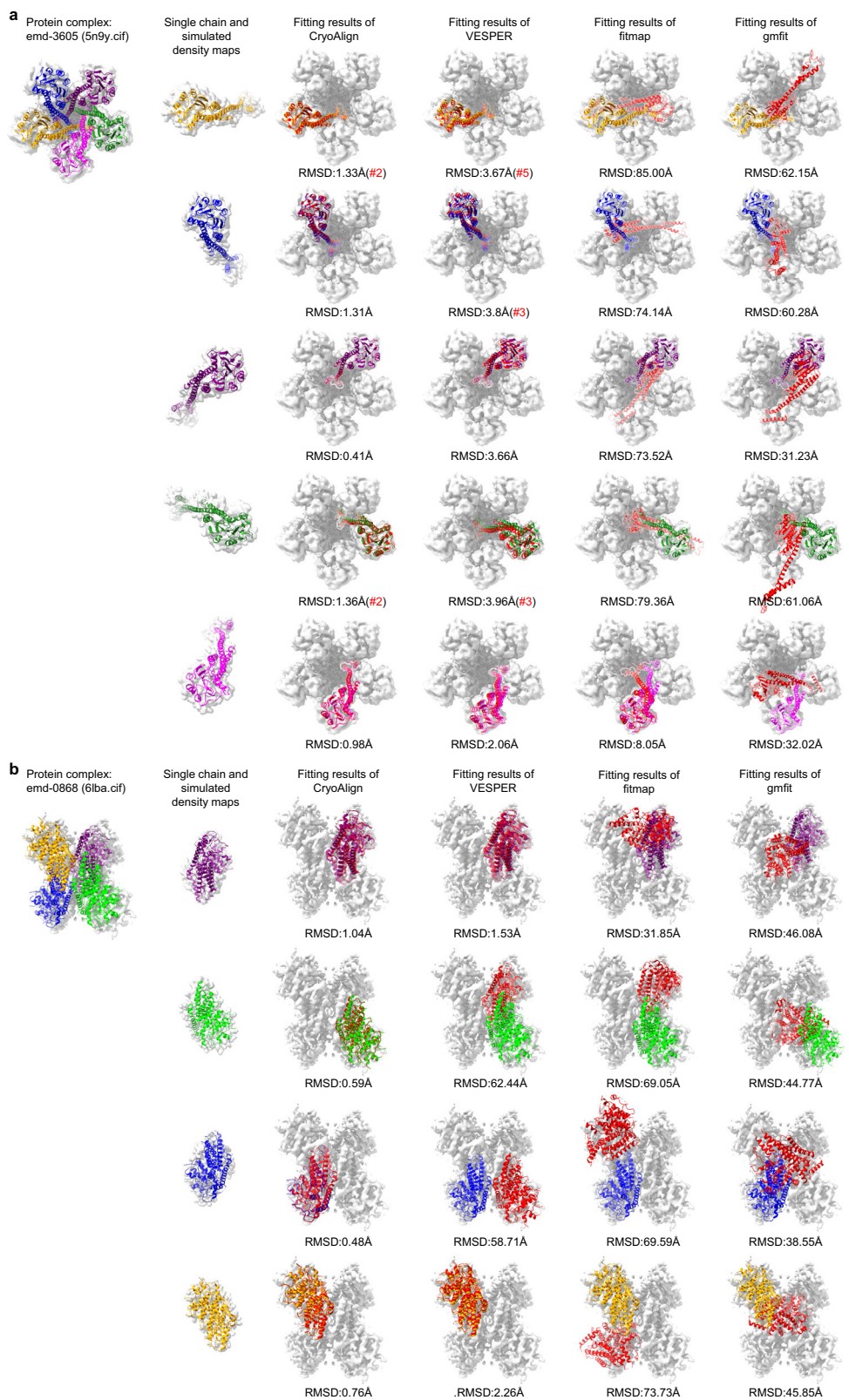

**Fig. 6 | Examples of atomic model fitting. a** Chain structure fitting of *pentameric ZntB transporter* (EMD-3605, PDB:5n9y), which consists of five single chains A, B, C, D and E. For rotational invariance caused by similarity of chains, we collected the five top-scoring results of CryoAlign or VESPER as candidates and selected the best one. The red "#2" beside the RMSD value represents the ranking of the best superimposition in the candidate list. **b** Chain structure fitting of *kinase domain-like (MLKL)* protein (EMD-0868, PDB:6lba), which consists of four single chains A, B, C and D. Note that if the RMSD value is large but no ranking is listed for CryoAlign or VESPER, none of the five top-scoring parameters resulted in a successful alignment.

points according to the input distance $p$ and a bandwidth $\sigma$:

$$k(p) = \exp\left(-1.5\left|\frac{p}{\sigma}\right|^2\right) \qquad (2)$$

**Clustering-based key point and descriptor extraction.** In cryo-EM maps, the density value corresponds to the integration of density functions related to atoms, and regions with high density can be indicative of protein backbones. CryoAlign employs the mean shift algorithm, a nonparametric clustering method, to effectively identify these dense regions in the map. Different from the density vector generation, CryoAlign determines the local density maximum points by the convergent results of the following iteration:

$$y_i^{t+1} = \frac{\sum_{n=1}^{N} k(y_i^t - x_n)\Phi(x_n)x_n}{\sum_{n'=1}^{N} k(y_i^t - x_{n'})\Phi(x_{n'})}, \qquad (3)$$

To enhance the representation capability and reduce the size of the point cloud, CryoAlign incorporates the DBSCAN (density-based spatial clustering of applications with noise) algorithm[30]. This algorithm clusters points that are located within a specified threshold distance, typically equivalent to the sampling space. By applying DBSCAN, CryoAlign groups nearby points together, effectively reducing the redundancy and capturing the essential structural information in a more compact form. The remaining points serve as key points for subsequent alignment stages.

Based on identified key-points and initial points assigned with "density vectors", CryoAlign proceeds to calculate density-based signature of histograms of orientations (SHOT) feature descriptors[37] for each key point (see Section "Density-based SHOT descriptor calculation" in Supplementary Material). CryoAlign examines the local neighborhood points surrounding each key point to calculate the modified SHOT descriptors. The orientations of the assigned density vectors at these neighboring points are quantized into discrete bins, and a histogram is constructed to collect the distribution of these orientations. This histogram effectively summarizes the local geometric characteristics of the density map concisely and informatively.

**Two-stage alignment**
After the sampling and clustering stages, two input density maps are efficiently transformed into point clouds and corresponding key points, denoted as $\{S_i, S_i^{key}\}$ for source (moving) map, and $\{T_j, T_j^{key}\}$ for target (fixed) map. In the first stage of alignment, CryoAlign utilizes a feature-based approach to estimate the initial transformation parameters. This involves collecting the key points and their corresponding feature descriptors from both the source and target point clouds. To efficiently reduce the size of the candidate set, CryoAlign employs a bidirectional nearest point matching strategy. This strategy assigns a binary value, denoted as $\delta(i,j)$, to each pair of key points, indicating whether they should be considered as a potential match. When $\delta(i,j) = 1$, the corresponding feature pair between key point $S_i$ and key point $T_j$ is considered a valid match. In contrast, when $\delta(i,j) = 0$, it means that the corresponding feature pair is discarded. The feature matching process is performed by bidirectionally checking the nearest neighbors:

$$\delta(i,j) = NN\left(S_i^{key}, T_j^{key}\right) \wedge NN\left(T_j^{key}, S_i^{key}\right), \qquad (4)$$

where $NN(\cdot, \cdot)$ determines whether the latter point is the nearest one to the former point in the feature domain. In other words, CryoAlign compares the Euclidean distances between the feature descriptors of key point $S_i^{key}$ and all the feature descriptors of key points $\{T_j^{key}\}$ in the target point cloud, and select the one with smallest distance as the nearest neighbor. Given the filtered feature point correspondences $\{S_i^{key}, T_i^{key}\}_{i=1}^{M}$, truncated least squares estimation and semidefinite

relaxation (TEASER)[38] are used to estimate the initial rigid transformation parameters, by optimizing the following objective function:

$$\min_{R \in SO(3), t \in \mathbb{R}} \sum_{i=1}^{M} \min\left(\left|T_i^{key} - RS_i^{key} - t\right|^2, \epsilon^2\right), \qquad (5)$$

where $R$ is the $3 \times 3$ rotation matrix and $t$ is the 3D translation vector.

The feature-based method provides a rough initial superimposition, while the point-based method aims to align the point clouds more closely. Accounting for the different distributions of the point clouds, CryoAlign utilizes the sparse-icp algorithm[39] in the second stage. This algorithm replaces the L-2 norm with the L-p norm (where $p < 1$), allowing for a higher tolerance for outliers. Unlike the first stage, which focuses on key point pairs, in the second stage, CryoAlign considers the initial point pairs $\{S_i, T_i\}_{i=1}^{N}$ generated by the nearest neighbor algorithm in 3D space. The optimization function based on point correspondences is formulated as:

$$\min_{R \in SO(3), t \in \mathbb{R}} \sum_{i=1}^{N} \left|T_i^{key} - RS_i^{key} - t\right|^2 + I_{SO(3)}(R), \qquad (6)$$

where $p < 1$ and $I_{SO(3)}$ constraints for the rotation matrix $R$.

**Similarity measuring function**
The similarity measuring function in CryoAlign is based on the aligned point clouds. Once the point clouds are transformed using the estimated alignment parameters, they are effectively superimposed. The similarity between the transformed point clouds $\{S_i\}$ and $\{T_j\}$, is measured along with their corresponding density vectors $\{u_i\}$ and $v_j$:

$$Similarity(S,T) = (1 - D_{JS}(S|T)) * \frac{\sum_k^N I(u_k, v_k)}{N}, \qquad (7)$$

$$I(u_k, v_k) = \begin{cases} 1 & u_k * v_k > \epsilon \\ 0 & otherwise \end{cases}, \qquad (8)$$

where $D_{JS}(\cdot)$ is the Jensen-Shannon divergence, measuring the global similarity of the spatial distributions; $N$ in the denominator represents the number of overlapped point pairs; and $I(\cdot, \cdot)$ is an indicator function, evaluating whether the dot product of two vectors is greater than a predefined threshold $\epsilon$. Notably, in local alignment, the Jensen-Shannon divergence is discarded because the segmented maps under masking operations reflect less distinction in spatial distributions.

**Exploration of local spatial features**
The combinations of keypoint detectors and feature descriptors are indeed crucial for achieving fast and effective initial alignment. There are several popular combinations available, such as keypoint detectors: 3D Harris[40], 3D SIFT (scale-invariant feature transform[41,42]), ISS (intrinsic shape signatures[43]); feature descriptors: SHOT (signatures of histograms of orientations[37]), FPFH (fast point feature histograms[44]), PFH (point feature histograms[45]), 3DSC (3D shape context[46]), USC (unique shape context[47]), ROPS (rotational projection statistics[48]). These algorithms are all computed with the PCL library[42]. In the case of CryoAlign, density vectors are utilized as the geometry attribute for each point, replacing the commonly used surface normals in point cloud processing. Comparing density vectors with surface normals is also an important aspect.

In the section "Local spatial feature descriptors" of Supplementary Material, we comprehensively analyze the performances of the aforementioned combinations and compare them with the results of CryoAlign's approach on the global alignment dataset. The analysis includes evaluations of surface normals and density vectors for their orientation consistency, as measured by cosine distances between matched points. Meanwhile, the performance of point correspondence establishment was assessed for different combinations of keypoint detectors and

descriptors through metrics such as failure ratios and the proportion of correct feature matching. Different feature matching strategies, including direct nearest neighbor and mutual feature matching, were also tested for all combinations. Through our analysis, we affirm that density vectors provide a better representation of geometric attributes compared to surface normals. The clustering-based keypoint extraction method also demonstrates superior performance. These methods consider the physical meaning of density values, making them more applicable to density maps. To encode geometric attributes into a feature vector for each keypoint, we utilize the SHOT descriptor architecture, resulting in a 352-dimensional feature representation of local structures. The experiments detailed in the Supplementary Material demonstrate that the SHOT architecture exhibits robustness and accuracy, particularly when used in the mutual feature matching strategy.

### Mask strategy for local alignment

In the local alignment, we take a moving spherical mask strategy to segment large volumes simply. The moving mask $M$ is created by configuring parameters such as the radius $r$, center $c$ and step distance $d$. For this study, the $r$ and $c$ values were adjusted to cover the small volume, while the step $d$ was set as half of the radius. By uniformly moving the mask, a series of alignment results and their respective similarity scores are obtained. The effectiveness is demonstrated in Table 4. In fact, most masks are redundant, and the mask strategy can be enhanced with provided initial poses. For example, existing exhaustive search methods are employed in larger intervals to obtain approximate rotation and translation values. Then CryoAlign utilizes the spherical mask within small regions around the initial pose, thereby significantly reducing the subsequent searching scope. In the section "Initial mask localization" of Supplementary Material, we take the results of exhaustive search method VESPER as the initial state and analyze the alignment performance under different sampling intervals. We find that CryoAlign consistently achieves the high-precision alignment and the initial position of mask only affects the success rate.

### Reporting summary

Further information on research design is available in the Nature Portfolio Reporting Summary linked to this article.

## Data availability

The data that support this study are available from the corresponding authors upon request. The datasets of cryo-EM maps for global or local alignment are provided in Supplementary data. The cryo-EM maps and fitted PDB entries can be downloaded from EMDB and PDB, respectively. The source data underlying Figs. 2, 3a, b, 4a–d, and Supplementary Figs. S1, S2, S4, S5 are provided as Source Data files. Global and Local alignment data can also be found in the Source Data files. For the illustration, an example dataset and corresponding analysis code are available [https://github.com/HeracleBT/CryoAlign/tree/main/data/example_dataset]. Source data are provided with this paper.

## Code availability

The CryoAlign program is freely available for academic use [https://github.com/HeracleBT/CryoAlign].

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

## Acknowledgements

This research was supported by the National Key Research and Development Program of China [2021YFF0704300], the King Abdullah University of Science and Technology (KAUST) Office of Research Administration (ORA) under Award No URF/1/4352-01-01, FCC/1/1976-44-01, FCC/1/1976-45-01, REI/1/5234-01-01, REI/1/5414-01-01, REI/1/5289-01-01, REI/1/5404-01-01, the National Natural Science Foundation of China Projects Grant [62072280, 61932018, 62072441, T2225007 and 32241027], the Natural Science Foundation of Shandong Province ZR2023YQ057, the Natural Science Foundation of Ningxia Province 2023AAC05036, and the Instrument Improvement Funds of Shandong University Public Technology Platform, with the help of SDU's Biomedical Research Center for Structural Analysis.

## Author contributions

X.G. and R.H. conceived the project and supervised the research. B.H. developed the methodology, performed the experiments, and analyzed the data. B.H. and R.H. designed the experiments. B.H. and F.Z. organized and wrote the paper. C.F. and J.Y. helped to revise the paper and provide scientific discussion when this study encountered problems.

## Competing interests

The authors declare no competing interests.
