## [Peer Review File · Nature Communications]

Accurate global and local 3D alignment of cryo-EM density maps using local spatial structural featuresReviewer #1 (Remarks to the Author):

The authors proposed a new algorithm to rapidly and accurately align EM density maps using local spatial structural features. It is a novel and innovative method in the field, mainly through the clustering-based key point detection and density-based local feature description. Overall, the paper is well organized, clearly written, and the method is described in details, along with the comprehensive and convincing experiments. The experimental results show the superiority of the method over the state-of-the-arts. It is expected that the method has important applications and will become an influential tool in the field.

There are few comments and suggestions that the authors should address in the revision, to further polish the quality of the paper.

Major:

1) For point cloud generation, the authors used the uniform sampling method. How about other sampling methods? The density maps of one protein may be acquired in different imaging parameters. Will the point clouds be affected by the imaging parameters, and so as the subsequent alignment?

2) The authors mentioned "key points ... reduce in size to around 10%~20% ..." in Section 2.3. But in Supplementary Materials, the authors said that other key point methods detect too few points. What is the appropriate number of key points? Should the number of key points be as high as possible or as low as possible?

3) The authors adopted 5Å as the sampling interval and used key point detection to reduce the size. Now that the clustering method can represent the backbone as the authors said, how about directly using the clustered points in 1~3Å(<5Å) sampling as the initial point cloud inputs? Additionally, why the second alignment was applied on the initial point clouds, not the extracted key points?

4) In the local alignment, the author proposed a simple translational mask strategy. Was the mask automatically generated? Please provide the details. If the mask is necessary, have the authors considered to take the result of exhaustive search method in the large intervals as the initial pose?

5) In the application in atomic model fitting, some complexes have symmetries, especially in Figure 7b. Did the authors take into account their symmetries when they calculated RMSD? The authors should provide more information about the rank. For example, the fifth ranked result of VESPER was provided in the Fig, how about the first ranked result? Why was it abandoned?

6) Compared to exhaustive search methods, the parameter setting seems to be more complicated. For better alignment results, the exhaustive search methods simply reduce the search intervals. What should users do for CryoAlign, if they want the higher alignment accuracy? The authors should list the parameters that can be modified and demonstrate their impacts.

Modifications:

1) In Figure 1, "Sample points..." should be "Point sampling...", to be consistent with other steps.

2) The subfigures e, f in Figure 4 are too large.

Reviewer #2 (Remarks to the Author):

The manuscript by He et al describes an alignment method for EM density maps. The EM density maps obtained after cryo-EM experiments are often reconstructed from tens or hundreds of thousands of single-particle images. They are of high signal-to-noise ratio (SNR) and presents little difficulty in 3D alignment. While the authors demonstrate the advantages of the proposed methods as to those existing methods like fitmap in Chimera, such improvements do not appear to represent a significant

advance. This is partially reflected in the moderate importance of the 3D alignment problem under high SNR conditions. 3D alignment is a useful, but often optional, tool in a cryo-EM study. Although I am certain that the as-described method will be of some use in the EM community, I do not believe that this work will be crucial for advancing the field and its importance matches those expected for a journal like Nature Communication. I therefore do not recommend its publication in this journal. When prepared for submission to elsewhere, the authors may consider the following issues for improvement of the manuscript.

- (1) Test the performance of the method under low SNR (for example, 0.1).
- (2) It is unclear if and how the alignment results are affected by the initialization of the procedure. Can one start from any orientation for 3D alignment by this method?
- (3) When implemented for atomic model fitting, does it support flexible fitting? If not, can you comment on the limitation of the method?
- (4) Can the method be adapted for computing 3D variance map for improved precision?
- (5) Many highly relevant references were not mentioned or cited. Please consider a more appropriate discussion and add relevant citations in the introduction and discussion.

Response to reviewers:

We are grateful to the two reviewers for their thoughtful and thorough comments, which helped us greatly improve our paper. We revised the paper following their comments. Below please find the point-by-point response to all the reviewers' comments, where the number index indicates the original comments (in blue color) and "Ans:" indicates our answers (in black color). The changes in the text in the Manuscript and Supplementary Material are marked in red.

=====
Reviewer 1

Comments to the Author

The authors proposed a new algorithm to rapidly and accurately align cryo-EM density maps using local spatial structural features. It is a novel and innovative method in the field, mainly through the clustering-based key point detection and density-based local feature description. Overall, the paper is well organized, clearly written, and the method is described in details, along with the comprehensive and convincing experiments. The experimental results show the superiority of the method over the state-of-the-arts. It is expected that the method has important applications and will become an influential tool in the field.

Ans: Thank you for the positive feedback and constructive comments. We followed all of them to improve the quality of the paper further. Please find the point-to-point response below to your comments.

Major:

1) For point cloud generation, the authors used the uniform sampling method. How about other sampling methods? The density maps of one protein may be acquired in different imaging parameters. Will the point clouds be affected by the imaging parameters, and so as the subsequent alignment?

Ans: Thank you for the great questions. We test other sampling methods such as topology representing networks (Martinetz and Schulten, 1994) and Poisson sampling on the global alignment dataset (Table 1). For comparison, the number of sampling points is set the same as that in uniform sampling. In fact, the initial point sampling does not significantly affect the registration results for input maps of similar resolutions. However, for "Cross resolution" maps, the TRN and Poisson sampling strategies show higher failure ratios. This is possibly due to the missing resolution information in the sampling process, or some parameters need to be adjusted, because we used only the common parameters in ProDy (Zhang *et al.*, 2021). In contrast, the sampling interval in uniform sampling ensures the resolution of extracted point clouds.

Table 1: Alignment evaluation in global dataset

Res. range	Uniform sampling(Å)/failure	TRN(Å)/failure	Poisson sampling(Å)/failure
<5Å	1.69/18.4%	1.78/20.0%	1.84/17.1%
5.0~10.0Å	2.88/6.25%	2.49/12.5%	2.69/12.5%
Cross res.	2.23/0%	2.32/23.08%	2.16/23.08%

There are two metrics calculated in the alignment evaluation, average RMSD and failure ratio. For RMSD, the smaller value means better alignment accuracy; for the failure ratio, the smaller value indicates higher stability.

Different imaging parameters definitely influence the density maps. Our method works on the post-processing of these volumes, thus the main influencing factors are the physical voxel sizes and the author recommended contour levels. We assume that the provided information is credible. The voxel size determines the uniform sampling interval, and the contour level aims to keep valid information. If the contour threshold is set too loose, the generated point cloud will have high noise.

2) The authors mentioned "key points . . . reduce in size to around 10% 20% . . ." in Section 2.3. But in Supplementary Materials, the authors said that other key point methods detect too few points. What is the appropriate number of key

points? Should the number of key points be as high as possible or as low as possible?

Ans: Thank you for the question. Actually the number of key points is just the basis of feature matching. The most crucial factor is the number of correctly matched key point pairs. Too many points make the matching process difficult and time-consuming to establish correct correspondences, due to the large candidate sets. For example, the average precision of feature matching between the two initial point clouds is only 17%, which is significantly lower than our current solution, which has achieved 29.2% precision. Too few points always lead to missing corresponding points. For example, in Supplementary Material Figure S2, the ISS detector extracts only tens of key points. If the feature description is with high quality, the feature matching would be extremely easy. That is why the combination of 3DSC and ISS has the highest precision (0.5) in Supplementary Material Table S2. However, the feature description is not always entirely correct, and key point sets with too few points lack the robustness to handle it. That is why the combination of 3DSC and ISS also achieves the highest failure ratio (98.4%). For rigid transformation estimation, at least dozens or hundreds of point pairs are needed. In the subsequent feature matching, many pairs will be filtered. As a rule of thumb, we suggest the number of remaining matching pairs be larger than 20 for subsequent parameter estimation.

3) The authors adopted 5Å as the sampling interval and used key point detection to reduce the size. Now that the clustering method can represent the backbone as the authors said, how about directly using the clustered points in 1 3Å(< 5Å) sampling as the initial point cloud inputs? Additionally, why the second alignment was applied on the initial point clouds, not the extracted key points?

Ans: Thank you for the comments. Although the clustered points can be roughly seen as the backbone representation, it discards considerable useful information. The clustered points, such as the key point in the local feature description, are more suitable as the bridge for information aggregation. We test acquiring the clustered points in 2Å as the initial points (the second column in Table 2), and the number of extracted points is approximately half of the one in 5Å uniform sampling.

Table 2: Alignment evaluation in global dataset

Res. range	Uniform sampling ¹ (Å)/failure	Clustering as initial ² (Å)/failure	ICP on key points(Å)/failure
<5Å	1.69/18.4%	2.13/14.29%	1.72/18.4%
5.0~10.0Å	2.88/6.25%	2.85/12.5%	2.93/6.25%
Cross res.	2.23/0%	3.75/30.77%	4.49/7.70%

¹ We take 5Å as the sampling interval in uniform sampling. ² We first sample points in 2Å, and then extract clustering centers as the initial points.

Applying the second alignment on the extracted key points is feasible (the third column in Table 2). However, the second-stage alignment relies heavily on the spatial distance of corresponding point pairs. For clustering processing, the extracted key points show an uneven distribution in the 3D space, leading to the less accuracy in RMSD, especially in the “Cross res” cases.

4) In the local alignment, the author proposed a simple translational mask strategy. Was the mask automatically generated? Please provide the details. If the mask is necessary, have the authors considered to take the result of exhaustive search method in the large intervals as the initial pose?

Ans: Yes, the mask is automatically generated. The moving mask is a sphere with three main parameters, center, radius and moving distances. The details are added in the section “Mask strategy for local alignment”. It is a good idea to perform local alignment with a given initial pose, such as the results provided by exhaustive search methods. This can significantly reduce computational costs, because our approach searches only a small area around the given pose. Certainly, our approach cannot handle the situation under low quality initial poses. We added the corresponding results in the main text section “Mask strategy for local alignment”.

“In the local alignment, we take a moving spherical mask strategy to segment large volumes simply. The moving mask M is created by configuring parameters such as the radius r , center c and step distance d . For this study, the r

and c values were adjusted to cover the small volume, while the step d was set as half of the radius. By uniformly moving the mask, a series of alignment results and their respective similarity scores are obtained. The effectiveness is demonstrated in Table 4.

In fact, most masks are redundant, and the mask strategy can be enhanced with provided initial poses. For example, existing exhaustive search methods are employed in larger intervals to obtain approximate rotation and translation values. Then CryoAlign utilizes the spherical mask within small regions around the initial pose, thereby significantly reducing the subsequent searching scope. Table 6 summarizes the average RMSD and failure ratios for initial pose generation at different sampling intervals. The initial alignment results were calculated using the exhaustive method VESPER, and the CryoAlign refinement results were acquired by searching $\pm r$ ranges around the initial ones. Due to the large sampling intervals, a failure threshold of 30Å is applied in this experiment. With intervals exceeding 10Å, even if the initial poses are imprecise, CryoAlign demonstrates the capability to rectify most of them by exploring the surrounding regions. Compared to the resolution range “<5Å” in Table 4, a decrease in the failure ratio is evident, indicating that a well-positioned mask contributes to improved alignment success. This improvement is achieved by masking feature points of large volumes, which interfere with feature matching processes. However, CryoAlign may not fully exploit well-positioned initial poses within the straightforward spherical mask strategy, since rotation and translation statistically have little effect on the feature construction.”

Table 3: Alignment evaluation with given initial pose in local alignment

	Sampling intervals for initial exhaustive			
	5Å	10Å	15Å	20Å
Initial exhaustive				
5Å	6.07/0%	10.26/6.56%	15.57/72.13%	7.76/94.26%
5.0~10.0Å	6.48/14.29%	7.71/14.29%	13.55/21.43%	19.45/78.57%
Cross res.	7.02/13.8%	11.10/23.08%	18.47/61.54%	21.01/90.77%
CryoAlign refinement				
5Å	4.34/5.74%	3.93/19.67%	3.99/21.31%	4.12/23.77%
5.0~10.0Å	2.89/7.14%	3.13/7.14%	3.42/7.14%	3.38/7.14%
Cross res.	5.20/13.8%	4.47/18.46%	4.77/21.53%	4.27/27.69%

There are two metrics calculated in the alignment evaluation, average RMSD and failure ratio. For RMSD, the smaller value means better alignment accuracy; for the failure ratio, the smaller value indicates higher stability.

5) In the application in atomic model fitting, some complexes have symmetries, especially in Figure 7b. Did the authors take into account their symmetries when they calculated RMSD? The authors should provide more information about the rank. For example, the fifth ranked result of VESPER was provided in the Fig, how about the first ranked result? Why was it abandoned?

Ans: Thank you for the suggestion. For atomic model fitting, it is common to fit one chain and then mask the corresponding regions. In the section “application in atomic model fitting”, we want to show the fitting ability without masking operations, which is why we provide the ranking information. In the Supplementary Material section “Ranking results in atomic model fitting”, we offered the top-ranking alignment results of the second example in the main text Figure 7.

“Many macromolecular complexes have multiple protein chains with very similar shapes. The precise alignment of a specific single chain which has similar “brother” chains raises a challenge to rank the exact transformation as high as possible in a number of alternative solutions. In the main text, we provide ranking information based on the optimal matching results. In this section, we enumerate the remaining alignments according to the ranking information and subsequently calculate RMSD values by determining the most suitable chain in Figure S8. It is evident that all candidate chains exhibit high structural similarity, nearly identical. The alignment of a single chain actually corresponds to multiple optimal parameters, the main reason for the low rankings of correct alignment. However, besides those optimal alignments, VESPER may provide multiple sub-optimal alignment results for the same target region, as observed in the rank#3 and rank#4 results of chain A. The consistent relative low accuracy of

VESPER assigns these specious alignment results approximate similarity scores and similar rankings. If the precision of correct alignment cannot stand out, these results may significantly affect its ranking. Compared to VESPER, CryoAlign delivers more accurate superimposition, with the RMSD values around 1.3Å. This high precision ensures the non-existence of sub-optimal alignments for the same target region in the top-ranking results.”

6) Compared to exhaustive search methods, the parameter setting seems to be more complicated. For better alignment results, the exhaustive search methods simply reduce the search intervals. What should users do for CryoAlign, if they want the higher alignment accuracy? The authors should list the parameters that can be modified and demonstrate their impacts.

Ans: Thank you for the good suggestion. We added the parameter settings to the Supplementary Material.

“In the generation of the initial point cloud, the primary parameter is the voxel sampling interval. In this study, we utilized 5Å as the default setting. This choice is suitable for density maps with resolution higher than 10Å, ensuring a sufficient number of points while maintaining acceptable execution times. Notably, if the input map represents a single chain or has small volumes, the sampling interval is better to be adjusted to 2 ~ 3Å for improved performance. It is crucial to ensure that the number of initial sampling points remains above 200. Then in the feature calculation and feature-based alignment processes, there are many parameters for the radius and weighting coefficients. We strongly recommend researchers follow the default settings, however for those requiring further details, interested readers can refer to Table S3.”

Modifications:

1) In Figure 1, “Sample points. . .” should be “Point sampling. . .”, to be consistent with other steps.

Ans: Thank you. We have corrected this.

2) The subfigures e, f in Figure 4 are too large.

Ans: Thank you. We have revised it.

For better presentation, we have reviewed the text thoroughly and corrected the grammar.

=====

Reviewer 2

Comments to the Author

The manuscript by He et al describes an alignment method for EM density maps. The EM density maps obtained after cryo-EM experiments are often reconstructed from tens or hundreds of thousands of single-particle images. They are of high signal-to-noise ratio (SNR) and presents little difficulty in 3D alignment. While the authors demonstrate the advantages of the proposed methods as to those existing methods like fitmap in Chimera, such improvements do not appear to represent a significant advance. This is partially reflected in the moderate importance of the 3D alignment problem under high SNR conditions. 3D alignment is a useful, but often optional, tool in a cryo-EM study. Although I am certain that the as-described method will be of some use in the EM community, I do not believe that this work will be crucial for advancing the field and its importance matches those expected for a journal like Nature Communication. I therefore do not recommend its publication in this journal. When prepared for submission to elsewhere, the authors may consider the following issues for improvement of the manuscript.

Ans: Thank you for the constructive comments. We have followed all of them to further improve the quality of the paper. Please find the point-to-point response below to your comments.

First and foremost, we would like to emphasize the fact that, current cryo-EM processing has been divided into two main stages: the signal and image processing stage that concentrates on the inverse problem from the 2D image signals to the 3D density map as you correctly pointed out; and the cryo-EM analysis and mining step that aims to compare, analyze and refine the reconstructed cryo-EM density map, and further mine the valuable information from the large number of processed density maps for downstream tasks, such as functional analysis, model building and drug development.

Our work aims to advance the latter one, post-processing of density maps, rather than reconstruction. For real/predicted protein PDB databases with scales reaching hundreds of millions, effectively searching and mining valuable information has become a prominent issue, as discussed in *Nat. Biotechnol.* (van Kempen *et al.*, 2023; Hamamsy *et al.*, 2023; Bileschi *et al.*, 2022), *Nat. Methods* (Linares-López *et al.*, 2023; Buchfink *et al.*, 2021), *Nat. Commun.* (Koehler Leman *et al.*, 2023) and *Nature* (Barrio-Hernandez *et al.*, 2023). Similarly, the same problem is emerging in the EMDB database, which currently holds more than 30,000 entries, with a fourfold increase since 2018. Obviously, with the popularity and maturity of EM-technologies, there will be a data explosion in the EMDB database in the near future. Many valuable works are exploring this problem in the cryo-EM field, offering their solutions, and contributing to top-tier journals. For example, VESPER, (Han *et al.*, 2021) as introduced in *Nat. Commun.*, is designed to align density maps and rank results using a scoring function, facilitating the mining of similar maps in the database. EMReady (He *et al.*, 2023), presented in *Nat. Commun.*, collects all EM entries at 3.0-6.0Å resolutions to explore the quality improvement scheme. Moreover, combined with real structures or predicted structures from the PDB database, several works such as EMBuild (He *et al.*, 2022), MAINMAST (Terashi and Kihara, 2018) in *Nat. Commun.*, DEMO-EM (Zhou *et al.*, 2022) in *Nat. Comput. Sci.* and CryoREAD (Wang *et al.*, 2023) in *Nat. Methods* have been developed to model PDB structures directly from density maps. Even within individual cryo-EM datasets, techniques such as difference map-based clustering and classification are adopted in works (Sheng *et al.*, 2023) published in *Nat. Commun.*, (Davis *et al.*, 2016) in *Cell* and (Dong *et al.*, 2023) in *Nucleic Acids Res.*, helping to identify more conformations and establish energy landscapes. It is clear that map alignment and comparison are either the main focus of the aforementioned studies, or serves as the fundamental step in these works.

With the explosion of density maps, existing tools face challenges in automatically providing a high-precision solution. This is the original motivation for our proposed scheme. Our method introduces a novel density map alignment architecture, using point cloud representation to extract local spatial features. Point clouds preserve the shape characteristics of the density maps while reducing a large number of calculations. By establishing point correspondences in the feature domain rather than the spatial domain, our approach has no requirements for initial poses, eliminating the reliance on manual intervention. Meanwhile, estimating rigid transformation parameters based on point correspondences enables our method to achieve sub-voxel accuracy. By performing automatic and accurate alignment, our method is anticipated to contribute to fast data mining through superimposition-based similarity, promotes conformation discovery and structure retrieval via large-scale map comparison, and supports high-precision protein chain assembly for PDB structure modeling. Comprehensive experiments in our paper demonstrate that our method significantly outperforms gmfit, fitmap, and VESPER in terms of both global and local alignment accuracy.

1) Test the performance of the method under low SNR (for example, 0.1).

Ans: Thank you for the great suggestion. Our method is designed specifically for the post-processing of cryo-EM density maps, characterized by high resolution and high SNR. Nevertheless, our method also exhibits a certain degree of robustness to noise, such as SNR=1.0, where noise can be filtered using point cloud clustering. Notably, an extremely low SNR such as 0.1 is common in subtomogram alignment, the key procedure for the resolution of the averaged one, and is the main procedure in 2D projection to 3D reconstruction. Generally, the alignment of subtomogram volume, which is with ~0.1 SNR and “missing wedge”, is taken by Fourier space cross-correlation considering CTF effects. Our method is not designed for subtomogram volume alignment but for the detailed analysis and data mining of cryo-EM density maps.

Nevertheless, following your comment, to illustrate the robustness, below we simulated the low SNR situations (1.0 and 0.1) using the example (EMD-3695, EMD-3696 main text Figure 1) in Figure 1. The density map is visualized in Chimera, employing the three-sigma values as the contour level, as in the point sampling process. As shown in Figure 1a, when SNR=0.1, determining the contour level becomes extremely challenging, and the structural signals are impossible to extract in the point cloud forms. However, upon increasing the SNR to 1.0, which is still low compared to the scenario that we targeted, the sampling points can depict the molecular shape, despite considerable noise. This is the basis for subsequent structural information extraction according to point cloud clustering. However, some default parameters should be modified. For example, simultaneously reducing the “distance_tolerance” and increasing the “clustersize_tolerance” in DBSCAN help to filter out clusters with only a few points. As illustrated in Figure 1b, key points extracted under default parameters are full of noise, making molecular shape recognition difficult. However, under the adjusted parameters, the extracted key-points are sparse but clean, rendering the molecular shape clearly

visible. It is noteworthy that this adjustment not only eliminates most noise but also filters points containing valid structural information. The alignment can then be accurately performed as in the high SNR situation.

Figure 1: **The example (EMD-3695, EMD-3696) for low SNR.** **a** The maps are visualized in Chimera using the three-sigma values as the contour level under different SNRs. Based on the contour level, the points are sampled in the 5Å interval. **b** When SNR=1.0, the key points are extracted in different clustering parameters. For filtering noises, we restrict the merging of clusters and eliminate small ones in DBSCAN. Then the sparse but clean key points can achieve a good alignment.

Furthermore, we implemented the adjusted parameters on the simulated global dataset (SNR=1.0) to assess alignment performance. Table 4 summarizes the average RMSD values and failure ratios for the compared methods. Fitmap shows a high failure ratio in the low SNR cases because it directly measures the incredible density correlation as the alignment criteria. VESPER preserves good stability as an exhaustive search method, due to its similarity measurement based on overall density vectors. Compared to density values, the trend of density values is less sensitive to noise. Similarly, CryoAlign collects the overall density vectors in a local region to depict structural characteristics. This statistical information effectively eliminates the interference from noise. As discussed in the main text, the utilization of local spatial features improves the alignment accuracy compared to global similarity-based schemes. Even in the low SNR cases, the advantage of local features still exists.

2) It is unclear if and how the alignment results are affected by the initialization of the procedure. Can one start from any orientation for 3D alignment by this method?

Ans: Thank you for the excellent question. We tested the performance under different initial poses by rotating density

Table 4: Alignment evaluation in simulated global dataset SNR=1.0

Res. range (Count)	CryoAlign(Å)/failure	VESPER(Å)/failure	fitmap(Å)/failure
<5Å(35)	1.94/28.57%	3.91/51.43%	0.97/71.43%
5.0~10.0Å(16)	2.64/25%	4.20/25%	0.55/50%
Cross res. (13)	3.38/23.08%	4.62/23.08%	3.02/69.23%

maps around various axes. Our method exhibits strong robustness to variations in initial poses. In this experiment, maps are rotated in three forms: around a single axis (x-, y-, and z-axes individually), around two axes ((x, y), (y, z), (x, z) respectively) and around all three axes (x, y, z simultaneously). Figure 2 offers some visual examples after given rotations.

Figure 3 presents the distribution of RMSD values in successful alignments. The similar colored regions suggest that initial rotations have negligible influence on accuracy. This robustness stems from the rotation invariance of the local spatial feature architecture when the local reference frame is precisely constructed. In real cases, factors such as noise and estimated surface normals can influence the frame establishment. However, the subsequent block-based statistical histograms further enhance the robustness. Table 5 summarizes the average RMSD and failure ratio. The consistent failure ratio implies that initial poses also have little impact on feature matching. Constructing local features is grounded in spherical areas around key points, and rotations do not significantly alter the count of adjacent points. Thus, to improve alignment success ratio, the primary focus should be on optimizing feature construction rather than initial poses.

Figure 2: **Some rotation examples** The volumes in the top row rotate around the z axis, while the second row around the (y, z) axes and the third row around all three axes. The rotation degrees are shown in (x, y, z) forms above the volumes.

Figure 3: **Accuracy of successful alignment in the rotated global dataset.** The colored regions represent the estimated RMSD distribution. Notably, the regions below zero hold no meaning; they are merely the result of distribution estimation.

Table 5: Alignment accuracy in rotated global dataset

Single axis	Res. range	60°(Å)/failure	120°(Å)/failure	180°(Å)/failure
	<5Å	1.58/17.14%	1.77/17.14%	1.62/17.14%
	5.0~10.0Å	3.20/12.50%	3.13/12.50%	3.12/12.50%
	Cross res.	2.35/0%	2.45/0%	2.49/0%
Two axes	Res. range	60°(Å)/failure	120°(Å)/failure	180°(Å)/failure
	<5Å	1.69/17.14%	1.72/17.14%	1.61/17.14%
	5.0~10.0Å	3.10/12.50%	3.14/12.50%	3.10/12.50%
	Cross res.	2.55/0%	2.32/0%	2.41/0%
Three axes	Res. range	60°(Å)/failure	120°(Å)/failure	180°(Å)/failure
	<5Å	1.84/17.14%	1.79/17.14%	1.73/17.14%
	5.0~10.0Å	3.14/12.50%	3.07/12.50%	3.14/12.50%
	Cross res.	2.20/0%	2.41/0%	2.21/0%

3) When implemented for atomic model fitting, does it support flexible fitting? If not, can you comment on the limitation of the method?

Ans: Thank you for bringing this up. We apologize for not emphasizing this important point in the original submission: our method achieves accurate rigid transformation to acquire well-assembled results, providing an important basis for flexible fitting. We added this clarification in the main text section “Application in atomic fitting” as follows:

“Moreover, through accurate rigid transformations, multiple chains are all placed into appropriate positions, serving as the initial assembling model. This well-assembled initial model is a crucial foundation for subsequent flexible fitting, an indispensable step in high-precision atomic modeling. CroAlign can conveniently integrate with existing point cloud-based approaches (Zampogiannis *et al.*, 2019; Hirose, 2021; Zhang *et al.*, 2018) to address this requirement. A protein structure typically consists of multiple chains. First, in CryoAlign, each chain is transformed into a point cloud, and aligned to the fixed map. CryoAlign transforms these point clouds representing chains respectively and merges them into a comprehensive and larger point cloud. The assembly of point clouds is an initial model representation of the protein structure. Then, the integrated point cloud as a whole, is compared to the fixed reference to estimate displacements for each point. Finally, the motion of point clouds can coherently translated into the atomic coordinates, as both point clouds and atoms share the same coordinate system. Interested researchers can refer to the Supplementary Material section “Extended results in flexible fitting” for the visual examples.”

The point cloud representation can directly describe the atomic model, serving as a bridge between atoms and density voxels. This representation puts atomic models and density maps into the same scale by uniform sampling, and constructs connections between backbone points extracted by structure-based clustering. We added an example and analysis in the Supplementary Material section “Extended results in flexible fitting”.

Figure 4: **An example for flexible transformation in atomic model fitting.** **a** The source map is EMD-4775, whose two chains A and B are extracted. The target map is EMD-4776. **b** The alignment results of individual chains and corresponding assembly results. **c** Based on the point clouds assembled by the compared methods, flexible fitting is applied to estimate displacements between matching points. Furthermore, the estimated displacement is performed on the atomic model to generate transformed PDB structure. The RMSD values are individually calculated for two chains.

“With the help of uniform sampling and structure-based clustering, the point cloud representation effectively bridges the gap between atoms and density voxels. Given the accurate rigid alignment, the flexible fitting, also called nonrigid registration in the point cloud, can be easily estimated by considering the displacements for matched point pairs. Many methods (Zampogiannis *et al.*, 2019; Hirose, 2021; Zheng and Doermann, 2006; Ma *et al.*, 2018; Zhang *et al.*, 2018) have been proposed for flexible fitting. In CryoAlign, we select the Bayesian coherent point drift algorithm (Hirose, 2021) to estimate displacements for each point. To illustrate the performance, we considered two conformations of the heterodimeric ABC exporter TmrAB (EMD-4775, EMD-4776). Fixing EMD-4776 as the reference map, we extracted chains A and B from EMD-4775 as assembling candidates, as depicted in Figure S9a, where the colored point clouds are attached nearby. Fitmap, lacking initial poses from experienced researchers, struggles to assemble

the two chains properly. Both CryoAlign and VESPER place the chains into the appropriate positions according to accurate local alignment. Then, these point clouds representing chains are transformed respectively and merged into a comprehensive and larger point cloud. The assembly of point clouds is an initial model representation of the protein structure (see Figure S9b). Fitmap provides a poor initial assembly model, making flexible fitting incorrectly reflect the molecular motion. In contrast, both CryoAlign and VESPER offer well-assembled point clouds, resulting in successful displacement estimation between matched points, as shown in Figure S9c. These point displacements can be conveniently applied to atomic models, as both points and atoms share the same coordinate system. Furthermore, compared to VESPER, CryoAlign achieves better fitting and lower RMSD, with a refinement of 0.1Å and 0.9Å for chains A and B, respectively. This is primarily because CryoAlign offers superior rigid alignment, thereby simplifying the establishment of point correspondences during the flexible fitting process. Notably, regions undergoing main conformational changes are magnified for enhanced visualization.”

Figure 5: **Another example for flexible transformation in atomic model fitting.** The source map is EMD-3198, whose chains A, B, C and D are extracted. The target map is EMD-3201. All chains are individually aligned by CryoAlign to the target map and assembled into a larger complex. Furthermore, the point displacements are estimated by flexible fitting to transform the corresponding PDB structure. The subunit with a large rotation is precisely depicted.

“In Figure S10, we present an additional example of the *E. coli* replicative DNA polymerase complex bound to DNA (EMD-3198, EMD-3201). The primary conformational distinction involves a large rotation in the chain A subunit,

highlighted within the black box. Based on the accurate assembly provided by CryoAlign, displacement construction between point pairs becomes straightforward. This transformation is further validated by the resulting PDB structure.”

4) Can the method be adapted for computing 3D variance map for improved precision?

Ans: Thank you for the question. For the example bL17-limited ribosome assembly intermediates in the main text, we collected a total of 42 different states from EMPAIR-10841 to compute the 3D variance map. We added a Figure and corresponding analysis in the main text section “Application in map comparison”.

Figure 6: **3D variance map of 42 different states of bL17-limited ribosome assembly intermediates.** Some representative ribosome assembly intermediates of different states are selected in the top row. The 3D variance map is displayed in the central slice of the yz plane, xy plane, and xz plane for visualization. The color intensities correspond to the variance values, with brighter colors indicating higher variances.

“Additionally, the high-precision alignment of CryoAlign enables an accurate map comparison of compared maps and helps the user more easily locate the variable regions and further analyze the conformational change. We collected a dataset of in total 42 different states of bL17-limited ribosome assembly intermediates from EMPAIR 10841. The 3D variance map was computed by fixing EMD-24491 as the reference map and aligning the remaining 41 conformations to it. Some examples of different states are presented in the top row of Figure 6. Notably, fitmap occasionally encounters alignment failures, as illustrated in Figure 5b. Consequently, the resulting variance images exhibit a uniform numerical distribution lacking differentiation, impeding the observation of conformational changes. VESPER delivers alignment results, albeit with less accuracy, facilitating the rough identification of variable regions with higher variances in the range [20, 30]. For example, the discernible changes in the upper parts of maps are apparent through analysis of the variance image in the y-z plane. However, the relatively lower alignment accuracy of VESPER introduces potential confusion between variable and stable areas, as variances in some stable regions also fall within the range [15, 20]. In contrast, the variance slice generated by CryoAlign reveals more pronounced distinctions between variable and stable regions. Here, larger variance values are concentrated in the range [20, 35], while smaller ones predominate in the range [0, 10]. These distinguished variance differences are the key to locating the conformational

changes and moving regions.”

5) Many highly relevant references were not mentioned or cited. Please consider a more appropriate discussion and add relevant citations in the introduction and discussion.

Ans: Thank you for the suggestion. We added some references in the main text section “Background” and “Application in atomic model fitting”:

“Moreover, with the advancement of cryo-EM technology, most of the recently solved cryo-EM structures have high resolution, ranging from 2Å to 10Å. Many important works (Herreros *et al.*, 2023; Chen and Ludtke, 2021; Zhong *et al.*, 2021) explore the continuous conformation changes to reconstruct a series of high-resolution maps, sufficiently enriching and characterizing the landscape of molecular states. All these factors indicate the coming of a high-resolution and big-data cryo-EM era. To extract and interpret the underlying structural information from cryo-EM density maps, there is a strong demand for accurate alignment and comparison of cryo-EM maps, especially for entries with high resolution. For example, comparison of superimposed density maps helps to identify variable areas associated with heterogeneity and to integrate 3D classification to establish conformational landscapes (Poitevin *et al.*, 2020; Rheinberger *et al.*, 2018; Joseph *et al.*, 2020; Dashti *et al.*, 2020; Dong *et al.*, 2023; Sheng *et al.*, 2023; Haselbach *et al.*, 2017). In protein macromolecular complex modeling, accurate local alignment effectively accelerates the chain assembly process (He *et al.*, 2022; Woetzel *et al.*, 2011; Zhang *et al.*, 2022; Van Zundert *et al.*, 2015), as the density of a subunit structure is simulated to find the best matching regions in experimental maps (Garzón *et al.*, 2007; Rossmann *et al.*, 2001; Nicholls *et al.*, 2018; De la Rosa-Trevin *et al.*, 2016).”

“Moreover, through accurate rigid transformations, multiple chains are all placed into appropriate positions, serving as the initial assembling model. This well-assembled initial model is a crucial foundation for subsequent flexible fitting, an indispensable step in high-precision atomic modeling. CroAlign can conveniently integrate with existing point cloud-based approaches (Zampogiannis *et al.*, 2019; Hirose, 2021; Zhang *et al.*, 2018) to address this requirement.”

We also added some references in the section “Extended results in flexible fitting” of Supplementary Material:

“With the help of uniform sampling and structure-based clustering, the point cloud representation effectively bridges the gap between atoms and density voxels. Given the accurate rigid alignment, the flexible fitting, also called nonrigid registration in the point cloud, can be easily estimated by considering the displacements for matched point pairs. Many methods (Zampogiannis *et al.*, 2019; Hirose, 2021; Zheng and Doermann, 2006; Ma *et al.*, 2018; Zhang *et al.*, 2018) have been proposed for flexible fitting.”

References

- Barrio-Hernandez, Yeo, J., Jänes, J., Mirdita, M., Gilchrist, C. L. M., Wein, T., Varadi, M., Velankar, S., Beltrao, P., and Steinegger, M. (2023). Clustering predicted structures at the scale of the known protein universe. *Nature*, pages 637–645.
- Bileschi, M. L., Belanger, D., Bryant, D. H., Sanderson, T., Carter, B., Sculley, D., Bateman, A., DePristo, M. A., and Colwell, L. J. (2022). Using deep learning to annotate the protein universe. *Nature Biotechnology*, **40**(6), 932–937.
- Buchfink, B., Reuter, K., and Drost, H.-G. (2021). Sensitive protein alignments at tree-of-life scale using diamond. *Nature methods*, **18**(4), 366–368.
- Chen, M. and Ludtke, S. J. (2021). Deep learning-based mixed-dimensional gaussian mixture model for characterizing variability in cryo-em. *Nature methods*, **18**(8), 930–936.
- Dashti, A., Mashayekhi, G., Shekhar, M., Ben Hail, D., Salah, S., Schwander, P., des Georges, A., Singharoy, A., Frank, J., and Ourmazd, A. (2020). Retrieving functional pathways of biomolecules from single-particle snapshots. *Nature communications*, **11**(1), 4734.
- Davis, J. H., Tan, Y. Z., Carragher, B., Potter, C. S., Lyumkis, D., and Williamson, J. R. (2016). Modular assembly of the bacterial large ribosomal subunit. *Cell*, **167**(6), 1610–1622.
- De la Rosa-Trevin, J. M., Quintana, A., Del Cano, L., Zaldívar, A., Foche, I., Gutiérrez, J., Gómez-Blanco, J., Burguet-Castell, J., Cuenca-Alba, J., Abrishami, V., *et al.* (2016). Scipion: A software framework toward integration, reproducibility and validation in 3d electron microscopy. *Journal of structural biology*, **195**(1), 93–99.
- Dong, X., Doerfel, L. K., Sheng, K., Rabuck-Gibbons, J. N., Popova, A. M., Lyumkis, D., and Williamson, J. R. (2023). Near-physiological in vitro assembly of 50s ribosomes involves parallel pathways. *Nucleic Acids Research*, **51**(6), 2862–2876.
- Garzón, J. I., Kovacs, J., Abagyan, R., and Chacon, P. (2007). Adp_em: fast exhaustive multi-resolution docking for high-throughput coverage. *Bioinformatics*, **23**(4), 427–433.
- Hamamsy, T., Morton, J. T., Blackwell, R., Berenberg, D., Carriero, N., Gligorijevic, V., Strauss, C. E., Leman, J. K., Cho, K., and Bonneau, R. (2023). Protein remote homology detection and structural alignment using deep learning. *Nature biotechnology*, pages 1–11.
- Han, X., Terashi, G., Christoffer, C., Chen, S., and Kihara, D. (2021). Vesper: global and local cryo-em map alignment using local density vectors. *Nature communications*, **12**(1), 2090.

- Haselbach, D., Schrader, J., Lambrecht, F., Henneberg, F., Chari, A., and Stark, H. (2017). Long-range allosteric regulation of the human 26s proteasome by 20s proteasome-targeting cancer drugs. *Nature communications*, **8**(1), 15578.
- He, J., Lin, P., Chen, J., Cao, H., and Huang, S.-Y. (2022). Model building of protein complexes from intermediate-resolution cryo-em maps with deep learning-guided automatic assembly. *Nature Communications*, **13**(1), 4066.
- He, J., Li, T., and Huang, S.-Y. (2023). Improvement of cryo-em maps by simultaneous local and non-local deep learning. *Nature Communications*, **14**(1), 3217.
- Herreros, D., Lederman, R. R., Krieger, J. M., Jiménez-Moreno, A., Martínez, M., Myška, D., Strelak, D., Filipovic, J., Sorzano, C. O., and Carazo, J. M. (2023). Estimating conformational landscapes from cryo-em particles by 3d zernike polynomials. *Nature Communications*, **14**(1), 154.
- Hirose, O. (2021). A bayesian formulation of coherent point drift. *IEEE Transactions on Pattern Analysis and Machine Intelligence*, **43**(7), 2269–2286.
- Joseph, A. P., Lagerstedt, I., Jakobi, A., Burnley, T., Patwardhan, A., Topf, M., and Winn, M. (2020). Comparing cryo-em reconstructions and validating atomic model fit using difference maps. *Journal of chemical information and modeling*, **60**(5), 2552–2560.
- Koehler Leman, J., Szczerbiak, P., Renfrew, P. D., Gligorijevic, V., Berenberg, D., Vatanen, T., Taylor, B. C., Chandler, C., Janssen, S., Pataki, A., et al. (2023). Sequence-structure-function relationships in the microbial protein universe. *Nature communications*, **14**(1), 2351.
- Llinares-López, F., Berthet, Q., Blondel, M., Teboul, O., and Vert, J.-P. (2023). Deep embedding and alignment of protein sequences. *Nature Methods*, **20**(1), 104–111.
- Ma, J., Wu, J., Zhao, J., Jiang, J., Zhou, H., and Sheng, Q. Z. (2018). Nonrigid point set registration with robust transformation learning under manifold regularization. *IEEE transactions on neural networks and learning systems*, **30**(12), 3584–3597.
- Martinetz, T. and Schulten, K. (1994). Topology representing networks. *Neural Networks*, **7**(3), 507–522.
- Nicholls, R. A., Tykac, M., Kovalevskiy, O., and Murshudov, G. N. (2018). Current approaches for the fitting and refinement of atomic models into cryo-em maps using ccp-em. *Acta Crystallographica Section D: Structural Biology*, **74**(6), 492–505.
- Poitevin, F., Kushner, A., Li, X., and Dao Duc, K. (2020). Structural heterogeneities of the ribosome: new frontiers and opportunities for cryo-em. *Molecules*, **25**(18), 4262.
- Rheinberger, J., Gao, X., Schmidpeter, P. A., and Nimigean, C. M. (2018). Ligand discrimination and gating in cyclic nucleotide-gated ion channels from apo and partial agonist-bound cryo-em structures. *Elife*, **7**, e39775.
- Rossmann, M. G., Bernal, R., and Pletnev, S. V. (2001). Combining electron microscopic with x-ray crystallographic structures. *Journal of structural biology*, **136**(3), 190–200.
- Sheng, K., Li, N., Rabuck-Gibbons, J. N., Dong, X., Lyumkis, D., and Williamson, J. R. (2023). Assembly landscape for the bacterial large ribosomal subunit. *Nature Communications*, **14**(1), 5220.
- Terashi, G. and Kihara, D. (2018). De novo main-chain modeling for em maps using mainmast. *Nature communications*, **9**(1), 1618.
- van Kempen, M., Kim, S. S., Tumescheit, C., Mirdita, M., Lee, J., Gilchrist, C. L., Söding, J., and Steinegger, M. (2023). Fast and accurate protein structure search with foldseek. *Nature Biotechnology*, pages 1–4.
- Van Zundert, C., Bonvin, M., et al. (2015). Fast and sensitive rigid-body fitting into cryo-em density maps with powerfit. *AIMS Biophysics*, **2**(2), 73–87.
- Wang, X., Terashi, G., and Kihara, D. (2023). Cryoread: de novo structure modeling for nucleic acids in cryo-em maps using deep learning. *Nature Methods*, pages 1–9.
- Woetzel, N., Lindert, S., Stewart, P. L., and Meiler, J. (2011). Bcl:: Em-fit: rigid body fitting of atomic structures into density maps using geometric hashing and real space refinement. *Journal of structural biology*, **175**(3), 264–276.
- Zampogiannis, K., Fermüller, C., and Aloimonos, Y. (2019). Topology-aware non-rigid point cloud registration. *IEEE Transactions on Pattern Analysis and Machine Intelligence*, **43**(3), 1056–1069.
- Zhang, S., Yang, K., Yang, Y., Luo, Y., and Wei, Z. (2018). Non-rigid point set registration using dual-feature finite mixture model and global-local structural preservation. *Pattern Recognition*, **80**, 183–195.
- Zhang, S., Krieger, J. M., Zhang, Y., Kaya, C., Kaynak, B., Mikulska-Ruminska, K., Doruker, P., Li, H., and Bahar, I. (2021). Prody 2.0: increased scale and scope after 10 years of protein dynamics modelling with python. *Bioinformatics*, **37**(20), 3657–3659.
- Zhang, X., Zhang, B., Freddolino, P. L., and Zhang, Y. (2022). Cr-i-tasser: assemble protein structures from cryo-em density maps using deep convolutional neural networks. *Nature methods*, **19**(2), 195–204.
- Zheng, Y. and Doermann, D. (2006). Robust point matching for nonrigid shapes by preserving local neighborhood structures. *IEEE transactions on pattern analysis and machine intelligence*, **28**(4), 643–649.
- Zhong, E. D., Bepler, T., Berger, B., and Davis, J. H. (2021). Cryodrnn: reconstruction of heterogeneous cryo-em structures using neural networks. *Nature methods*, **18**(2), 176–185.
- Zhou, X., Li, Y., Zhang, C., Zheng, W., Zhang, G., and Zhang, Y. (2022). Progressive assembly of multi-domain protein structures from cryo-em density maps. *Nature computational science*, **2**(4), 265–275.

Reviewer #1 (Remarks to the Author):

The authors have comprehensively addressed my comments as well as the other reviewer comments. The topic is trendy and the methodology is updated. I recommend it to be accepted.

Reviewer #1 (Remarks on code availability):

All source code are available on that GitHub repository. It has Docker images for easy installation and running.

Reviewer #2 (Remarks to the Author):

The authors have addressed my previous questions.